# Assessing the impact of climate change on high return levels of peak flows in Bavaria applying the CRCM5 Large Ensemble

Florian Willkofer[1], Raul R. Wood[1,2,3], Ralf Ludwig[1]

[1] Department of Geography, Ludwig-Maximilians-Universität München, Munich, 80333, Germany
[2] WSL Instistute for Snow and Avalanche Research SLF, Davos Dorf, 7260, Switzerland
[3] Climate Change, Extremes and Natural Hazards in Alpine Regions Research Center CERC, Davos Dorf, 7260, Switzerland

*Correspondence to*: Florian Willkofer (florian.willkofer@campus.lmu.de)

**Abstract.** Severe floods with extreme return periods of 100 years and beyond have been observed in several large rivers in Bavaria in the last three decades. Flood protection structures are typically designed based on a 100-year event, relying on statistical extrapolations of relatively short observation time series while ignoring potential temporal non-stationarity. However, future precipitation projections indicate an increase in the frequency and intensity of extreme rainfall events, as well as a shift in seasonality. This study aims to examine the impact of climate change on the 100-year flood ($HF_{100}$) events on 98 hydrometric gauges within the Hydrological Bavaria. A hydrological climate change impact (CCI) modelling chain consisting of a regional single model initial condition large ensemble (SMILE) and a single hydrological model was created. The 50 equally probable members of the CRCM5-LE were used to drive the hydrological model WaSiM to create a hydro-SMILE. As a result, a database of 1,500 model years (50 members x 30 years) per investigated time period was established for extreme value analysis (EVA) to illustrate the benefit of the hydro-SMILE approach for a robust estimation of the $HF_{100}$ based on annual maxima (AM), and to examine the CCI on the frequency and magnitude of $HF_{100}$ in different discharge regimes under a strong emission scenario (RCP8.5). The results demonstrate that the hydro-SMILE approach provides a clear advantage for a robust estimation of the $HF_{100}$ using empirical probability on 1,500 AM compared to its estimation using the generalized extreme value (GEV) distribution on 1,000 samples of typically available time series size of 30, 100, and 200 years. Thereby, by applying the hydro-SMILE framework the uncertainty from statistical estimation can be reduced. The study highlights the added value of using hydrological SMILEs to project future flood return levels. The CCI on the $HF_{100}$ varies for different flow regimes, with snowmelt-driven catchments experiencing severe increases in frequency and magnitude, leading to unseen extremes that impact the distribution. Pluvial regimes show a lower intensification or even decline. The dynamics of $HF_{100}$ driving mechanisms depict a decline in snow melt driven events in favor of rainfall driven events, an increase of events

driven by convective rainfall, and almost no change of the ratio between single driver and compound events towards the end of the century.

## 1 Introduction

The devastating force of floods poses a significant threat to infrastructure, livestock, and human life. In Germany, two of the most severe floods in the last three decades were the 2002 and 2013 flood events (along with other

major events in 1999, 2005, and 2016) (Thieken et al., 2016; Blöschl et al., 2013). The 2002 and 2013 events caused a total of about 17 billion Euros in economic damage due to their large spatial extent and high water levels, with the 2013 flood considered the most extreme event in the last sixty years (Thieken et al., 2016). However, different climatic and catchment conditions caused these events, with the 2002 event resulting from intense rainfall leading to flash floods across multiple small catchments, and the 2013 event due to high antecedent soil moisture

from long-lasting precipitation followed by more moderate but spatially widespread rainfall (Thieken et al., 2016). In addition to precipitation magnitude, other flood drivers such as antecedent soil moisture conditions, snowmelt, as well as flood driving processes determined by catchment and river characteristics contribute to the non-linearity of the hydrological response to extreme precipitation events (Blöschl et al., 2015). Recent studies analyzing European flood events over the last five decades suggest an increase in the magnitude and frequency of high flows

and flood events depending on the event type and region (Blöschl et al., 2019; Bertola et al., 2020; Blöschl et al., 2015). However, this trend depends on the time frame considered for the analysis, and the evaluation period remains crucial for either the estimation or the development of high return periods (Blöschl et al., 2015; Schulz and Bernhardt, 2016). Precipitation (heavy precipitation and long-lasting rainfall) and snowmelt (in regions with snowmelt-governed regimes) remain the primary natural causes of flooding, with other influences (e.g., catchment

characteristics, antecedent catchment conditions, compound events with snow- or glacier melt) and snowmelt becoming less important once a certain threshold of extreme precipitation is exceeded (Brunner et al., 2021b). According to the sixth Intergovernmental Panel on Climate Change (IPCC) Assessment Report, there is high confidence that a warmer climate will intensify wet weather and climate conditions affecting flooding (IPCC, 2021). Even with a 1.5 °C warming limit under the Paris agreement, heavy precipitation, along with extreme

discharge events, is likely to intensify in Europe, with increasing confidence above 2 °C warming (IPCC, 2021). For most discharge gauges, observational records begin in the 19[th] century or even later (Blöschl et al., 2015). Although most of these observations offer sufficiently long time series of data for estimating peak flows of moderate return periods, they still hinder a robust statistical estimation of extreme return periods, such as the 100-year flood and above. These types of extreme hydrological events are required for structural flood protection and

risk management (Wilhelm et al., 2022; Brunner et al., 2021a; Blöschl et al., 2019). Brunner et al. (2021a) illustrate the challenges in modeling and predicting high flows due to data availability, process representation, and human influences.

Recently, SMILEs have emerged as a powerful tool to enhance statistical analysis of extremes in climatological behavior (von Trentini et al., 2020; Wood and Ludwig, 2020; Wood et al., 2021; Aalbers et al., 2018; Martel et
al., 2020). Unlike other common ensembles of different global or regional climate model (GCM/RCM) combinations, SMILEs comprise multiple equiprobable realizations (members) of a single GCM or GCM/RCM combination that differ only in their initial conditions, representing the chaotic nature of the climate system (Arora et al., 2011; Fyfe et al., 2017; Kirchmeier-Young et al., 2017; Sigmond et al., 2018; Leduc et al., 2019). The actual model structure, physics, parameterization, external forcings are preserved. Thus, SMILEs offer a profound
database for analyzing internal (or natural) climate variability (Wood and Ludwig, 2020; Martel et al., 2018), separating natural variability from an actual change signal (Aalbers et al., 2018; Wood and Ludwig, 2020), and extreme events (Wood et al., 2021; Martel et al., 2018). Applying SMILEs for hydrological modelling allows for the creation of a so-called hydro-SMILE, which in turn allows for the exploitation of vast data for the analysis of the hydrological response of catchments to extreme precipitation events.

Due to the high spatio-temporal resolution, this ensemble-based climate and hydrological modeling approach is computationally demanding. However, the high spatio-temporal resolution of a hydro-SMILE is particularly valuable for an enhanced representation of extreme values in models as it allows for spatially refined catchment features (e.g., slopes, soil characteristics, land use) and more precise values (e.g., discharge) due to higher temporal resolution. Thus, this study focuses on only a single region comprised by the major Bavarian river basins (upper
Danube, Main, Inn) with all their tributaries to account for the computational demand as well as the advantages gained by the high resolution.

In this study, a climatological SMILE is employed to drive a physically based hydrological model with high spatio-temporal resolution for the major Bavarian river catchments. The resulting hydro-SMILE is used to answer the following questions:

a) Is there a benefit applying a SMILE for hydrological impact modelling regarding the estimation of high flows of large return periods?

    b) How does climate change affect the dynamics in frequency and magnitude of extreme discharges?

    c) How are the driving mechanisms of these extreme discharges changing?

Although the data presented in this study would allow for an analysis of events beyond the 100-year flood, we
focus on this extreme event to answer these questions as this event is widely used in literature, higher return periods are prone to increased uncertainties (e.g., $HF_{1000}$), and it serves as design criterion for water management

infrastructure in this region and elsewhere. The study area is first introduced in section 2.1, followed by an overview of the climatological SMILE post-processing in section 2.2.1. The hydrological model setup used to produce the hydro-SMILE along with an evaluation of its performance are then presented in section 2.2.2. The subsequent sections describe the methods to illustrate the benefit of a hydro-SMILE for the estimation of peak flow with high return periods (section 2.2.3), to assess the influence of climate change on the change in magnitude and frequency of the 100-year flood (section 2.2.4), and to determine the changes in drivers of events with magnitudes of at least the 100-year flood (section 2.2.5). Finally, the results of the analysis are then presented in sections 3.1 to 3.3 and later discussed in section 4, followed by concluding remarks in section 5.

## 2 Study Area, Data, and Methods

### 2.1 Study Area

This study focuses on the major Bavarian rivers, including the upper Danube upstream of Achleiten, Main, Inn, and upstream tributaries of the Elbe, as well as their smaller and larger tributaries originating from adjacent states (Bade-Württemberg, Hessen, Thuringia) and countries (Austria, Switzerland, Italy, Czech Republic). The catchments of these rivers extend beyond the political borders of Bavaria (Figure 1). The entirety of these catchments is referred to as the Hydrological Bavaria in this study.

The Hydrological Bavaria covers approximately 100,000 km² and features a diverse landscape ranging from the Alps (with the highest point being Piz Bernina at 4049 meters above sea level; m.a.s.l) and the alpine foreland in the south to the southern German escarpment in the north of the study area (with the lowest point being 90 m.a.s.l at Frankfurt-Osthafen) and the eastern mountain ranges to the east (Willkofer et al., 2020; Poschlod et al., 2020). The complexity of these landscapes and different climatological conditions (up to 1100 mm annual total precipitation sums in the north, 2500 mm in the south; an mean annual temperature of 10 °C in the north, down to 5 °C (-8 °C on alpine summits; Poschlod et al. (2020)) in the south results in a variety of runoff regimes (Poschlod et al., 2020).

The discharge of many rivers within the Hydrological Bavaria is influenced by artificial retention structures (i.e., dams, retention basins), naturally formed lakes, or transfer systems (drinking water supply, low flow elevation) (Willkofer et al., 2020). The major river catchments were divided into a total of 98 smaller sub-catchments to better represent the various flow regime types of the respective gauges which are further of common interest for flood protection (Willkofer et al., 2020).

120

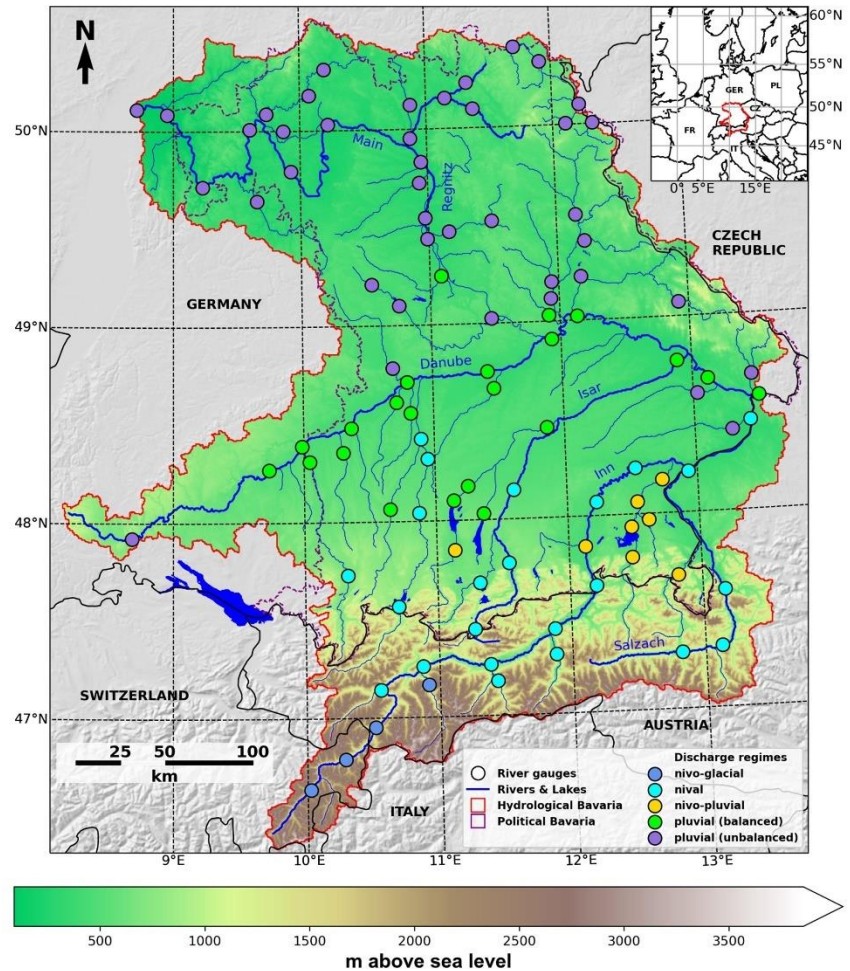

**Figure 1: Map showing the elevation of the Hydrological Bavaria (red line) which comprises the political Bavaria (dashed purple line) and the 98 hydrometric gauges used in this study as well as their respective discharge regime type (colored dots) at their respective rivers (blue lines).**

**2.2 Data and Methods**

To assess the impact of climate change on extreme return periods of peak flows, the hydroclimatic modeling chain illustrated in Figure 2 was introduced within the scope of the ClimEx project (Climate Change and Hydrological Extreme Events, www.climex-project.org). This common chain is divided into a climate and a hydrological impact section and covers three spatial scales (GCM scale, RCM scale, hydrological model scale) with increasing resolution along the chain.

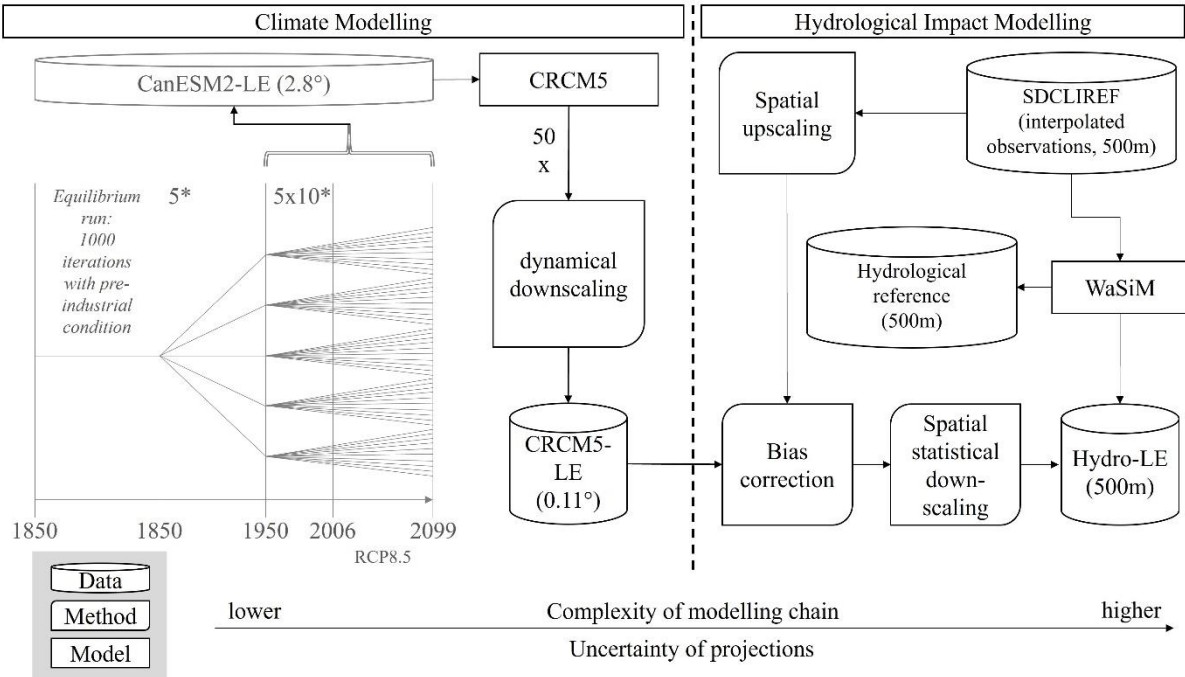

**Figure 2: The ClimEx modelling chain uses the CanESM2 large ensemble (LE, gray, not created within the ClimEx project) to generate the CRCM5-LE. The CRCM5-LE is then used to explore the impacts of climate change on the hydrology of the Hydrological Bavaria through a hydrological large ensemble (Hydro-LE) created using the hydrological model WaSiM. The SDCLIREF dataset of interpolated meteorological observations was employed to calibrate and validate the hydrological model as well as for the bias correction. The CRCM5-LE represents a SMILE, consisting of a single model that downscales output from the employed ESM using slight differences in the initialization.**

Since the introduced model chain requires a vast number of computational resources, the ClimEx project employed the high-performance computing systems of the Leibniz Supercomputing Centre (LRZ) as well as its technical and consultative support to migrate and adapt software and data to its systems, facilitate calculations, and provide an extensive amount of storage to archive the data and make them available to the scientific community (data available at https://www.climex-project.org).

### 2.2.1 Climate data

A SMILE composed of 50 independent members of the Canadian Earth System Model, version 2 (CanESM2) large ensemble (LE) was used as a base for all further analysis. The CanESM2-LE was produced by the Canadian Centre for Climate Modelling and Analysis (CCCma) and described in previous publications (Fyfe et al., 2017; Kirchmeier-Young et al., 2017; Arora et al., 2011; Leduc et al., 2019). All members of the CanESM2-LE used natural and anthropogenic forcings for the historical period from 1950 to 2005 and the representative concentration

pathway 8.5 (RCP8.5; van Vuuren et al., 2011) emission scenario from 2006 to 2099 (Kirchmeier-Young et al.,

2017; Leduc et al., 2019; Fyfe et al., 2017; Sigmond et al., 2018). The individual members differ only in their initial conditions rather than changes in model structure, physics, or parameters, and therefore offer a range of internal or natural variability of the climate system at a global scale.

These 50 members were dynamically downscaled from ~2.85° (≈ 310 km) to 0.11° (≈ 12 km) using the Canadian Regional Climate Model, version 5 (CRCM5; Martynov et al., 2013; Šeparović et al., 2013) over two spatial

domains, the European and the northeastern North American domains (Leduc et al., 2019). As with the CanESM2-LE, variations between the individual members were obtained by unique initial conditions for each member, thus providing a range of internal or natural variability on a regional scale. The resulting CRCM5 large ensemble (CRCM5-LE; Leduc et al., 2019) of 50 transient members provides the basis for assessing the impact of climate change on hydro-meteorological extreme events for the Hydrological Bavaria.  We focus on the model years 1961

to 2099 as opposed to 1950 to 2099 to account for the time it takes for the RCM to produce fully independent realizations due to the inertia of the ocean model (Leduc et al., 2019). A comparison between the CRCM5-LE and the E-OBS observational gridded dataset (Haylock et al., 2008) at the CRCM5 grid revealed biases for a historical period between 1980 and 2012, showing regional and seasonal variations in magnitude of temperature and precipitation over Europe (Leduc et al., 2019). Since the creation of RCM-LEs is challenging in terms of

computational demand (performance and storage) only a few are available to date (Addor and Fischer, 2015; Leduc et al., 2019; Aalbers et al., 2018; Brönnimann et al., 2018). However, all of the RCM-LEs differ in their domain size, spatial resolution and ensemble size. All RCM-LEs are ensembles of opportunity and are dependent on the availability of the driving GCM-LEs which in the CMIP5-phase are all based on RCP8.5. Thus, in this study only a single RCM-LE as well as a single scenario was employed.

Since this bias was considered to affect the behavior of the outputs of the hydrological model due to shifts in seasonality and magnitude, a bias correction was applied. The required meteorological data of precipitation, air temperature, relative air humidity, incoming shortwave radiation, and wind speed were adjusted to match a meteorological reference of interpolated 3-hourly station data (Sub-Daily Climate Reference, SDCLIREF; Ludwig et al., 2019) on the RCM grid using an adaptation of the quantile-mapping approach after Mpelasoka and Chiew

(2009). This approach as described in Willkofer et al. (2018) involved using multiplicative or additive correction factors, and was further adapted for using 3-hourly correction factors for every quantile and month (for further details, see S3). To preserve an internal spread between the members, a single set of factors was deduced from a combination of all 50 members. Despite the numerous benefits (increasing reliability of climate change projections of the hydrological impact model, reducing bias in mean annual discharge) and shortcomings (disrupting feedbacks

between fluxes, modification of change signals, assumption of a stationary bias) of bias correction    (e.g.,

Teutschbein and Seibert, 2012; Maraun, 2016; Ehret et al., 2012; Dettinger et al., 2004; Chen et al., 2021; Huang et al., 2014), bias correction is often inevitable for climate change impact studies (Gampe et al., 2019).

Subsequently, the bias corrected data were statistically downscaled to the hydrological model scale (500 m x 500 m) using a mass preserving approach (Marke, 2008; Ludwig et al., 2019). This approach involved the spatial interpolation (inverse distance weighting) of anomalies for each time step from the monthly mean reference state (1981-2010) at the CRCM5-LE cell center points to the hydrological model scale (Brunner et al., 2021b). The interpolated time step anomalies at the hydrological scale were then applied (multiplied or added) to the respective gridded monthly climatological reference fields of the SDCLIREF (Brunner et al., 2021b). In order to ensure the mass conservation, the downscaled RCM data was upscaled to the original RCM grid scale (mass conservative remapping) and compared to the RCM timestep values, to determine any correction factors necessary which were then applied to the downscaled grid cells to close the mass balance.

For further details, readers are referred to a comprehensive summary in the Supplementary Materials for the CanESM2-LE (S1), the CRCM5-LE (S2), and the bias correction (S3).

### 2.2.2 Hydrological Model WaSiM

The Water balance simulation Model (WaSiM; Schulla, 2021) was employed to perform the hydrological simulations driven by the CRCM5-LE resulting in a hydro-SMILE (the WaSiM-LE). WaSiM is a distributed, mostly physically-based, and deterministic model for simulations on various spatial (1 m to 10 km) and temporal (minute to daily) scales with a constant time step. It includes routines for evapotranspiration, snow accumulation and melt, glaciers, soil water transfer, groundwater, discharge generation and routing (Schulla, 2021). The model is frequently used for hydrological climate change impact studies for small-scale to mesoscale catchments on various topics, such as glaciers, groundwater, and discharge (Iacob et al., 2017; Neukum and Azzam, 2012; JÓNSDÓTTIR, 2008).

The model was set up in high spatio-temporal resolution (500 m and 3 h) for 98 catchments of the Hydrological Bavaria with a focus on high flow representation using distributed data derived from the European DEM (EU-DEM; European Environment Agency, 2013b), land use data provided by the CORINE land cover dataset (European Environment Agency, 2013a), distributed soil information from the European Soil Database (ESDBv2.0; European Environment Agency, 2013a), as well as groundwater information provided by the Hydrogeologische Übersichtskarte (HÜK; Dörhöfer et al., 2001) and IMHE (IHME; BGR, 2014). A single set of parameters for distributed parameters (i.e., evapotranspiration, soil properties) was defined globally for the entire modeling domain (Willkofer et al., 2020). Although there are abundant in situ data available for the study region, these are mainly provided as point measurements which are often representative of the entire catchment area and

therefore require interpolation to the hydrological model resolution which can introduce large uncertainties. Furthermore, some approaches of the hydrological model offer free parameters which cannot be measured. Hence, a calibration of the model for a limited number of free as well as usually locally measurable parameters was deemed necessary. For further details about the model calibration procedure we would like to refer to Willkofer et al. (2020). Local parameters for discharge storage components (i.e., interflow, direct flow) were calibrated using an automated algorithm (dynamically dimensioned search (Tolson and Shoemaker, 2007) and simulated annealing with progressing iterations (Černý, 1985; Kirkpatrick et al., 1983)) minimizing a weighted combination of performance metrics (overall metric OM; Eq. 1), including Nash and Sutcliff efficiency (NSE; Nash and Sutcliffe, 1970), Kling-Gupta efficiency (KGE; Gupta et al., 2009), the logarithmic NSE and the ratio of root mean squared error to standard deviation (RSR; Moriasi et al. (2007)) (Willkofer et al., 2020). A best fit would result in OM = 0, with larger deviations from 0 indicating a worse model fit. Due to the focus on high flow representation more emphasis was placed on the respective measures (i.e., NSE and KGE). For further details about the model setup the reader is referred to Willkofer et al. (2020).

$$OM = 0.5 \times (1 - NSE) + 0.25 \times (1 - KGE) + 0.15 \times (1 - logNSE) + 0.1 \times RSR \qquad (1)$$

The simulations of a single parameter set for various catchments within a heterogeneous landscape revealed satisfactory to very good results for most of the 98 gauges during the 30-year reference period of 1981 to 2010. However, for a few gauges, the model was not able to reproduce the observed discharge satisfactorily (16 (5) gauges showing NSE (KGE) values below 0.5, see also Figure S1 a and b in the supplements S4) (Willkofer et al., 2020; Poschlod et al., 2020). Furthermore, the simulations reproduced the mean high flow sufficiently well, with over 60% of the gauges showing absolute deviations from observed values below 20%. Nonetheless, gauges in alpine or pre-alpine catchments exhibited a deficit in mean high flow values due to the lack of observed precipitation resulting from an undercatch of precipitation for that region (Poschlod et al., 2020). Consequently, the level of trust (LOT) for peak flows of return periods of 5, 10, and 20 years flood events, introduced in Willkofer et al. (2020) showed a moderate to high confidence for most catchments. The LOT further depends on the model performance to a certain degree, where gauges depicting a lower model performance often exhibit a lower LOT as well. LOT were not provided for extreme flood events (i.e., 100-year flood events) since they are subject to significant epistemic uncertainty due to the restricted availability of simulated data (30 years). In Brunner et al. (2021a) the same hydrological model simulations were evaluated on a daily timescale, in contrast to the 3-hourly timescale here, in terms of general evaluation metrics (i.e., NSE, KGE, volume efficiency, and mean absolute error), as well as for flood specific characteristics (i.e., number of events, mean timing of the event, mean volume, mean duration). The evaluation of flood characteristics showed a good agreement on the number of events showing

only a slight underestimation of events, a good agreement on the timing of events with only a slight delay in flood occurrence, as well as an overestimation of flood volume and duration.

Due to the holistic calibration approach employing a single set of parameters over several heterogeneous catchments, the in parts poor performance of individual catchments can be expected, as catchment specific characteristics can only be considered to a certain extent (e.g., karstic soils, transfer systems, artificial reservoirs). While calibrating each catchment individually might lead to a higher performance at the respective catchment scale, it also increases the likelihood of overfitting. Furthermore, since the hydrological model serves for climate

change impact analysis, relative change values are of more interest than changes in absolute values. Nonetheless, the performance must be considered for interpretation. A brief overview of the model's performance for each gauge is given in the supplement materials (S4).

The resulting hydro-SMILE comprises 50 members of transient simulated data from 1961 to 2099, providing a total of 6,950 model years to be exploited to analyze extreme values.

**2.2.3 Benefit of a hydro-SMILE for the estimation of extreme peak flows**

This study used the simulated discharge for the reference period of 1981 to 2010 out of the entire dataset to assess the benefits of the hydro-SMILE in estimating return levels. Like the individual members of the CRCM5-LE, the members of the WaSiM-LE are equally probable and, therefore, provide a comprehensive database to facilitate the analysis of extreme values.

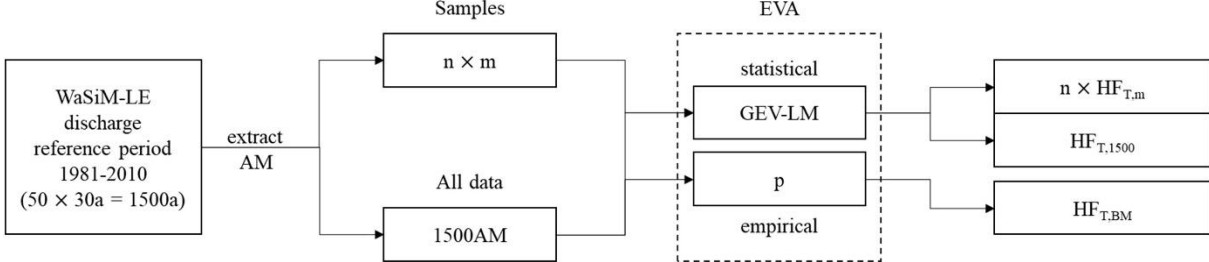

**Figure 3: Process chain illustrating the benefit of a hydro-SMILE for climate change impact studies on peak flows of extreme return periods. The process includes extreme value analysis (EVA) based on annual maximum (AM), with bootstrapping resampling to create n different samples of sample size (m). The probability of non-exceedance (p) and the Generalized Extreme Value (GEV) distribution with the L-Moments (LM) estimators are used to derive estimates for high flow values of the return period T (HF$_T$) for the samples (m), all data (1500), and the benchmark (BM). The statistical analysis was performed using the extRemes package (v2.0) for R (Gilleland and Katz, 2016).**

Figure 3 illustrates the approach taken to emphasize the benefits of the hydro-SMILE in analyzing peak flows of high return periods for the reference period. The 30-year reference period (ref) was selected for all 50 members, resulting in 1,500 model years (50 members x 30 years) of discharge data for each of the 98 gauges. First, the annual maximum of each model year (hydrological year) was extracted for the analysis. Since the database consists of 1,500 model years, this number is considered sufficient to employ empirical non-exceedance probabilities (Martel et al., 2020). However, to demonstrate the benefit of the hydro-SMILE database a statistical analysis using the stationary GEV distribution was also conducted for comparison purposes. A bootstrapping approach with resampling was used to create 1,000 samples ($n$) of different sizes $m$ (30, 100, 200) years (each sample without replacement). Using 1,000 samples ensures that one value of the 1,500 AM has a chance of >99% of being selected by chance for $m = 30$. The GEV was employed to estimate the return periods and corresponding confidence intervals. The parameters of the GEV distribution were estimated using L-Moments. The GEV distribution was selected as it is among the better performing methods relying on AM (Bezak et al., 2014) and is the recommended choice for German gauges (Salinas et al., 2014; Fischer and Schumann, 2016).

Although the sample size of 30 and 100 AM may be small for estimating peak flows of high return periods, they were selected along with a size of 200 AM as they represent an average (30 years) to rare (100 & 200 years) data availability of observed discharge values at different gauges (GRDC, 2021). The resulting 1,000 estimates for return levels of peak flows offer a comprehensive database to demonstrate the benefit of the hydro-SMILE. Additionally, the GEV was calculated using the entire 1,500 AM database for each gauge to allow for a comparison with a benchmark value. This benchmark for the return levels of peak discharge was deduced by applying the quantile based on the empirical probability of non-exceedance $p$ (Eq. (2)) to all 1,500 AM values for each gauge, and it is considered to represent a robust estimate. This analysis focused on the 100-year flood, which is an event

of a 100-year return period $T$ (HF$_{100}$; $T$ = 100) and the corresponding 99[th] percentile $p$ of the distribution of the 1,500 AM values as a benchmark.

$$p = 1 - \frac{1}{T} \tag{2}$$

Values for the benchmark derived by the empirical probability as well as the HF$_{100}$ values estimated using the GEV are further normalized to the benchmark to allow for a better comparison.

### 2.2.4 Projection of changes in frequency and magnitude

This study further investigates the dynamics of magnitude and frequency of the HF$_{100}$ for three future periods (near future: 2020-2049; mid future: 2040-2069; far future: 2070-2099) frequently used in similar CCI studies (e.g., Hattermann et al., 2018) and providing the same database of 1,500 AM values as for the benefit analysis. Therefore, the robust estimates of extreme return levels of peak flows derived by the empirical probabilities are used for the assessment of climate change impacts on their magnitude ($C_M$, Eq. (3)) and frequency ($C_F$, Eq. (4a to c)) in the three future periods.

$$C_M = \left( \frac{HF_{T_{fut}} - HF_{T_{ref}}}{HF_{T_{ref}}} \right) \cdot 100 \ \% \tag{3}$$

$$C_F = \frac{1}{1 - f\left(HF_{T_{ref}}\right)} \tag{4a}$$

$$f = F\left(HF_{T_{fut}}\right) \tag{4b}$$

$$F(x) = \sum_{i=1}^{j} h_i = \sum_{i=1}^{j} \frac{h(x_i)}{n} \tag{4c}$$

The change in magnitude is given as the difference between the future ($HF_{Tfut}$) and reference value ($HF_{Tref}$) relative to the reference value in percent. The change in frequency is expressed as the return period value $T$ and is calculated by applying the empirical cumulative distribution function $F$ (ECDF with frequency for an event $h_i$ described as the ratio between the frequency for the specific event $h_{(xi)}$ and the number of all values $n$, Eq. (4c)) for the respective future period ($f$, Eq. (4b)) to the value of the 100-year flood of the reference period (Eq. (4a)). The percentile value of f for the reference 100-year flood value is then used to deduce the future return period by solving the empirical probability of non-exceedance for the return period $T$ (Eq. (4a)). The change signals are calculated for each of the above mentioned 30-year future periods. However, this analysis requires stationarity for the underlying data. Since we use the entire 1,500 model years provided by the 50 members, we determine stationarity if less than 5 % of the members exhibit a significant trend for each individual gauge. A Mann-Kendall (MK) test (Mann, 1945; Kendall, 1955) for stationarity conducted on each individual member and gauge revealed no significant trend for the

reference period (with significance level $\alpha = 0.01$) for more than 95 % of the members along all gauges. However, for the future periods the MK test exhibits significant trends for more than 5 % of the members in 6 of the 98 gauges. Limiting the evaluation periods to 20 years instead of 30 years lead to similar results for the MK test showing no apparent trend for all gauges in the reference period, but showing for at least one gauge a significant

trend (more than 5 % of members with a trend) in the future periods. Studies by Poschlod et al. (2020) and Brunner et al. (2021b) conducted their analysis on the same database using time slices of at least 30 years as well. Thus, we choose to use 30-year periods since stationarity criteria are met in most catchments and opt for the larger database, as well as maintaining consistency with these studies.

### 2.2.5 Dynamics in driving mechanisms

The employed process-based hydrological model allows for a more detailed investigation of the dynamics in driving mechanisms of extreme discharges of the 100-year flood and beyond. First, extreme events of magnitudes of at least the $HF_{100}$ are extracted for each of the 98 gauges and 50 members of the hydro-LE. To avoid sampling a single event multiple times a 5-day period was used to separate individual events as suggested by Svensson et al. (2005). The starting date of the events served as entry to extract data from potential flood drivers which include

precipitation, melt from snow and glaciers, and soil water content prior to the event. Precipitation events were further separated into heavy rain events (hr) and steady rain events (sr), while there was no distinction between liquid and solid precipitation. The respective thresholds to identify and separate the different precipitation event types (see Table 1) were adopted from the German Weather Service (DWD, Deutscher Wetterdienst, 2024).

**Table 1: Thresholds for the identification of the driving mechanism (driver) of extreme discharge events above $HF_{100}$.**
**P represents the precipitation events heavy rain (hr) and steady rain (sr), Melt represents melting water from snow and glaciers (in mm snow water equivalent), and $SWC_{root}$ represents the soil water content of the soil's root zone.**

| Driver | Sub-category | Volume | Accumulation period |
|---|---|---|---|
| P | hr | 15 mm | 3 h |
| | hr | 20 mm | 6 h |
| | sr | 25 mm | 12 h |
| | sr | 30 mm | 24 h |
| | sr | 40 mm | 48 h |
| | sr | 60 mm | 72 h |
| Melt | snow | 15 mm | 2 weeks |
| | glacier | 15 mm | 2 weeks |
| $SWC_{root}$ | | 110% * $\mu(SWC_{root})_{REF}$ | 2 weeks |

A melt driven event (snow and/or glacier) was identified if the snow water equivalent from melt exceeded 15 mm prior two weeks of the event (adapted from Brunner et al., 2021b). Extreme discharge events may as well be caused

by a superposition of these driving mechanisms and are often referred to as compound events (e.g., rain on snow,

rain on saturated soils). Thus, if more than one driver is identified, we ascribe it to the compound event type. Furthermore, since an elevated soil water content cannot be responsible for an extreme discharge event alone, it is only considered as a contributing factor and is always part of the compound event type. In this case, the contribution of an elevated soil water content is considered when the soil water content is 10% higher than the long term mean soil water content for the entire reference period. All possible combinations (single drivers and compound events) result in 32 different or superimposed mechanisms. To reduce complexity and to focus on specific aspects of the different drivers we aggregate the 32 possible combinations to six different major contributions listed in Table 2.

**Table 2: Composition of the different drivers for the analysis of the dynamics of the HF$_{100}$ driving mechanisms.**

| | |
|---|---|
| **Melt vs. rainfall** | |
| Melt | only melt events and associated compounds - excluding rainfall compounds |
| Rainfall | all rainfall driven events and associated compounds - including compounds with melt events |
| **Heavy rainfall vs. steady rainfall** | |
| heavy rainfall | all event types which include heavy rain – including compounds with steady rainfall |
| steady rainfall | all event types which include steady rain – excluding any compounds with heavy rainfall |
| **Single vs. compound event** | |
| single event | all events caused by a single driver |
| compound event | all events caused by multiple drivers |

First, we analyze the dynamics of melt versus rainfall events, second the dynamics of heavy rainfall (convective events) versus steady rainfall (advective events) are investigated, and third we illustrate the dynamics of single drivers (any type) versus compound events (any type).

## 3 Results

### 3.1 Benefits of hydro-SMILEs for the estimation of extreme return periods of peak flows

Large ensembles provide a vast amount of data, therefore they are considered to be beneficial for extreme value analysis (Kendon et al., 2008; Kjellström et al., 2013; Wood and Ludwig, 2020). The benefit of a hydro-SMILE to determine robust extreme hydrological discharge values for Hydrological Bavaria are analyzed, specifically for the 100-year flood. The robust values for the discharge gauges, derived using the empirical probability of non-exceedance for a 100-year event, serve as a benchmark for comparison with values derived by the GEV distribution using three different sample sizes (30, 100, 200) of AM values (Figure 4).

The results shown in Figure 4 (a, b, and c) illustrate that the estimates of HF$_{100}$ are more robust with an increasing number of AM values used for the GEV, as indicated by the decreasing spread of the blue markers around the

black benchmark line with increasing sample size. Table 3 summarizes the statistical characteristics of the deviation of the estimates from the benchmark across all 98 gauges. While the range of the relative deviation of the 1,000 samples of $HF_{100}$ estimates from the benchmark is between 0.33 and 2.71 when calculated with a sample size of 30 AM values (Figure 4a), this range diminishes to 0.49 and 1.91 for 100 AM values (Figure 4 b) and 0.56 and 1.60 for 200 AM values (Figure 4 c). Therefore, the range of the 1,000 estimates diminishes with an increase in sample size and the values cluster more densely around the benchmark. However, despite the remaining non-negligible range of deviations from the benchmark, the mean (1.01) as well as the median (0.98 to 1.0) across all values for all gauges are close to the benchmark value for different sample sizes. The inner 50 % of the 1000 samples across all 98 gauges exhibit the largest deviation with a sample size of 30 AM (between 0.84 and 1.15) and the lowest for 200 AM (0.94 to 1.07). Therefore, only 25 percent of the samples show underestimations below 0.84 (0.92, 0.94) and only 75 percent exhibit larger overestimations than 1.15 (1.08, 1.07) with a sample size of 30 AM (100 AM, 200 AM). Thus, with deviations larger than 15 % for 50 percent of the estimates calculated using a sample size of 30 AM, only half of the estimated $HF_{100}$ values are within an acceptable range (±15 %, considering model parameter uncertainty and errors in observations affecting the model quality regarding high flows) compared to the benchmark. This number increases with a larger sample size.

**Table 3: Summary of overall statistics of the relative deviation of the $HF_{100}$ estimates from the benchmark value across all gauges. The table incldues the number of sample (n), sample size (m) given in annual maximum (AM) values and the 0.25/0.75 quantile (Q25, Q75) values.**

| N | m | minimum | Q25 | mean | median | Q75 | maximum |
|------|------|---------|------|------|--------|------|---------|
| 1000 | 30 | 0.33 | 0.84 | 1.01 | 0.98 | 1.15 | 2.71 |
| 1000 | 100 | 0.49 | 0.92 | 1.01 | 1.00 | 1.08 | 1.91 |
| 1000 | 200 | 0.56 | 0.94 | 1.01 | 1.00 | 1.07 | 1.60 |
| 1 | 1500 | 0.98 | 1.00 | 1.02 | 1.01 | 1.03 | 1.09 |

While the majority of gauges show estimates that are evenly distributed around the benchmark, some gauges exhibit a tendency towards over- or underestimation of the $HF_{100}$ estimates with more values falling above or below the benchmark line. This behavior may be different when using more than 1000 samples to conduct the analysis. The difference between the benchmark value obtained from empirical probability and the estimates obtained from the GEV distribution can vary greatly depending on the samples selected from 1,500 AM values.

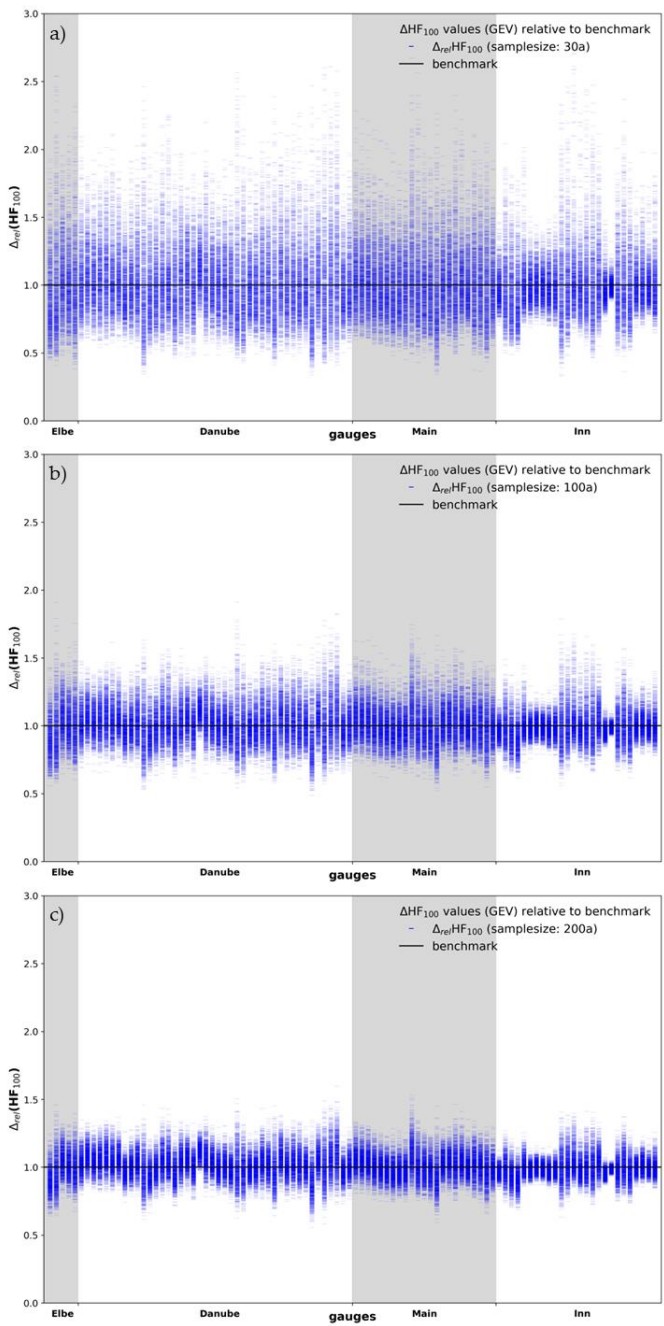

**Figure 4: Comparison of HF$_{100}$ estimates calculated using the GEV distribution with 1000 AM samples of a) 30, b) 100, and c) 200 years per gauge (blue markers) with the respective benchmark value (solid black line) for 98 gauges.**

A comparison between the $HF_{100}$ estimates derived using the empirical probability of non-exceedance and those obtained using the GEV distribution is shown in the supplements (S5). The values gained by the GEV distribution still exhibit deviations from the benchmark, although being marginally different from it.

**3.2 Changing dynamics of the 100-year peak flows in future projections**

The changes in $HF_{100}$ for the investigated gauges in Hydrological Bavaria in the 21$^{st}$ century are summarized for five distinct discharge regimes (defined by the Pardé coefficient) which were adapted from Poschlod et al. (2020) (Figure 1). The regimes comprise the glacio-nival regime of four high Alpine catchments, a nival regime of mostly Alpine to pre-Alpine catchments, a nivo-pluvial regime of pre-Alpine catchments, a balanced pluvial regime (little variation in mean monthly discharges) along the Danube and its tributaries in the Alpine foreland, and the unbalanced pluvial regime (more pronounced peak in monthly discharge from January to March) (Poschlod et al., 2020). One gauge that was originally assigned to its own regime in Poschlod et al. (2020) has been re-allocated to the pluvial (unbalanced) regime, as it exhibits a similar mean discharge behavior.

Within the study area, the flood protection structures are typically desinged based on a stipulated estimation of $HF_{100}$ from observations, which represent a stationary condition in the past. Any future increase in the magnitude and frequency of these extreme values poses a threat to these structures.

Figure 5 displays violin plots that illustrate the range of changes in the magnitude of $HF_{100}$ events for the different discharge regimes as well as the distribution of changes across the respective clusters of gauges for the near (horizon 2035), mid (horizon 2055), and far future (horizon 2085) periods. Overall, 78 % of all gauges (76/98) show an increase in magnitude for the 2035 horizon, 76 % (74/98) for the 2055 horizon, and 89 % (87/98) for the 2085 horizon.

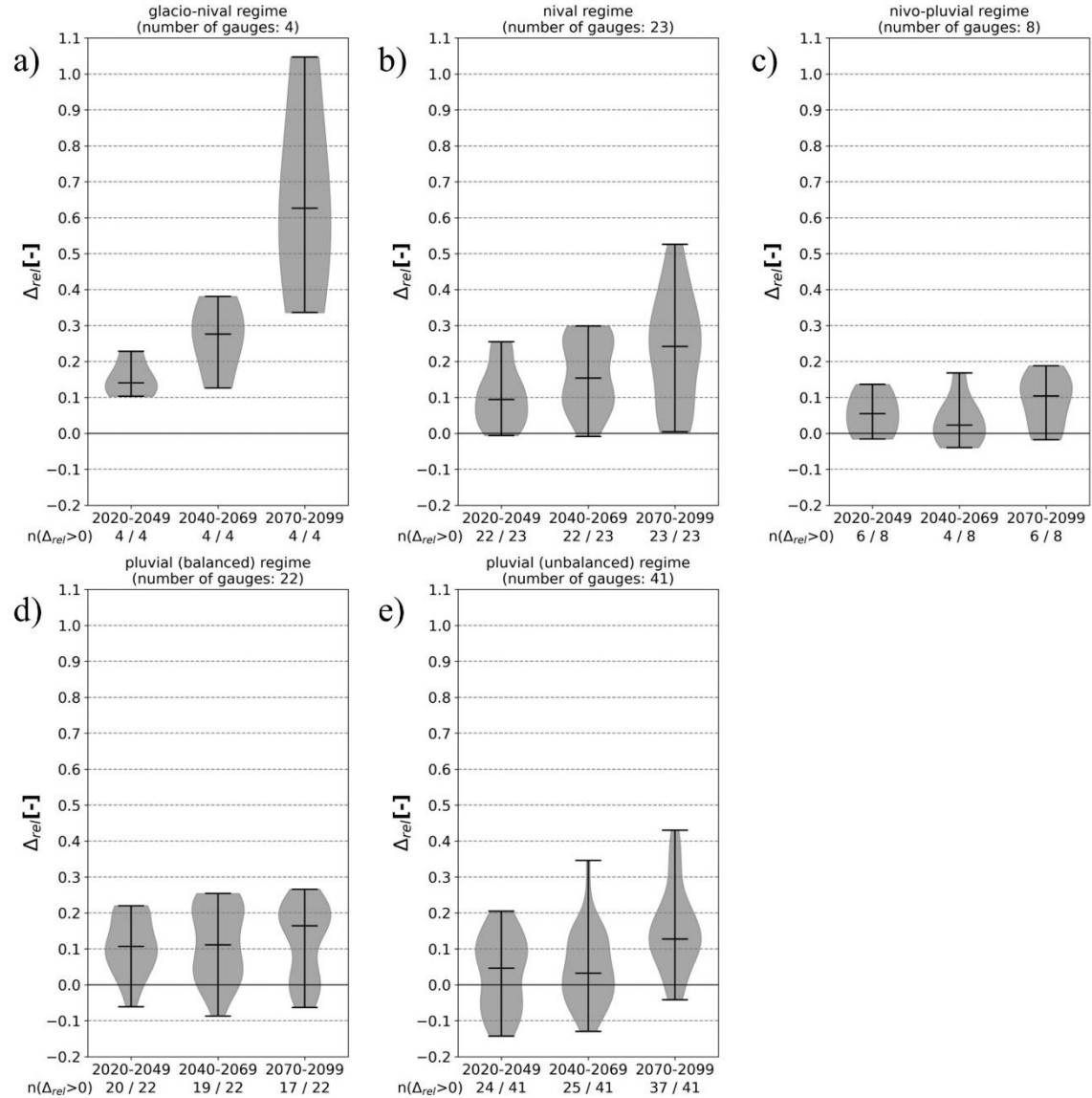

**Figure 5: Violin-plots indicating the changes of the magnitude of the $HF_{100}$ for the three future periods (near, mid, far) compared to the reference period, with changes presented as relative difference ($\Lambda_{rel}$) between the reference and the future $HF_{100}$ value for each gauge. Results of the 98 gauges are aggregated for the five discharge regimes (a = glacio-nival, b = nival, c = nivo-pluvial, d = pluvial (balanced), e = pluvial (unbalanced)). The figures display the total number of gauges per regime as well as the number of gauges depicting an increase in magnitude.**

The CCI are most severe for the glacio-nival regime (Figure 6a), as all three future periods exhibit an increase in magnitude of the $HF_{100}$ events of at least 10% compared to the reference period. The nivo-pluvial regime (Figure 6c) shows the smallest spread and the lowest increase in $HF_{100}$ magnitude across all future periods compared to

the reference period. As the distance from the Alps increases and the discharge regimes shift from snowmelt influenced to more precipitation driven, the number of gauges projecting a decrease in $HF_{100}$ intensities increases. However, the majority of gauges still exhibit an increase in intensities, with up to 18.8% for the nivo-pluvial Figure 5c), 26.6% for the balanced pluvial (Figure 5d), and 43 % for the unbalanced pluvial regime (Figure 5e) in the far

future. The gradient of an increase in magnitude over all three projection periods is small for the nivo-pluvial and balanced pluvial regimes, which show the least intensification of $HF_{100}$ values for the respective periods. However, the gradient of increase is more distinct for the remaining regimes, with the largest increase in the glacio-nival regime (Figure 5a). The gauges in this regime depict the strongest increase in $HF_{100}$ intensities for the 2085 horizon, with an increase of 36.6 % to 104.7 %.

Based on the future projections of the hydro-SMILE, the discharge values of the $HF_{100}$ are likely to increase for most of the gauges of Hydrological Bavaria. Consequently, the frequency of the $HF_{100}$ discharge for the reference period also increases. Figure 6 shows the change in frequency between the future and the reference period for the different regimes.

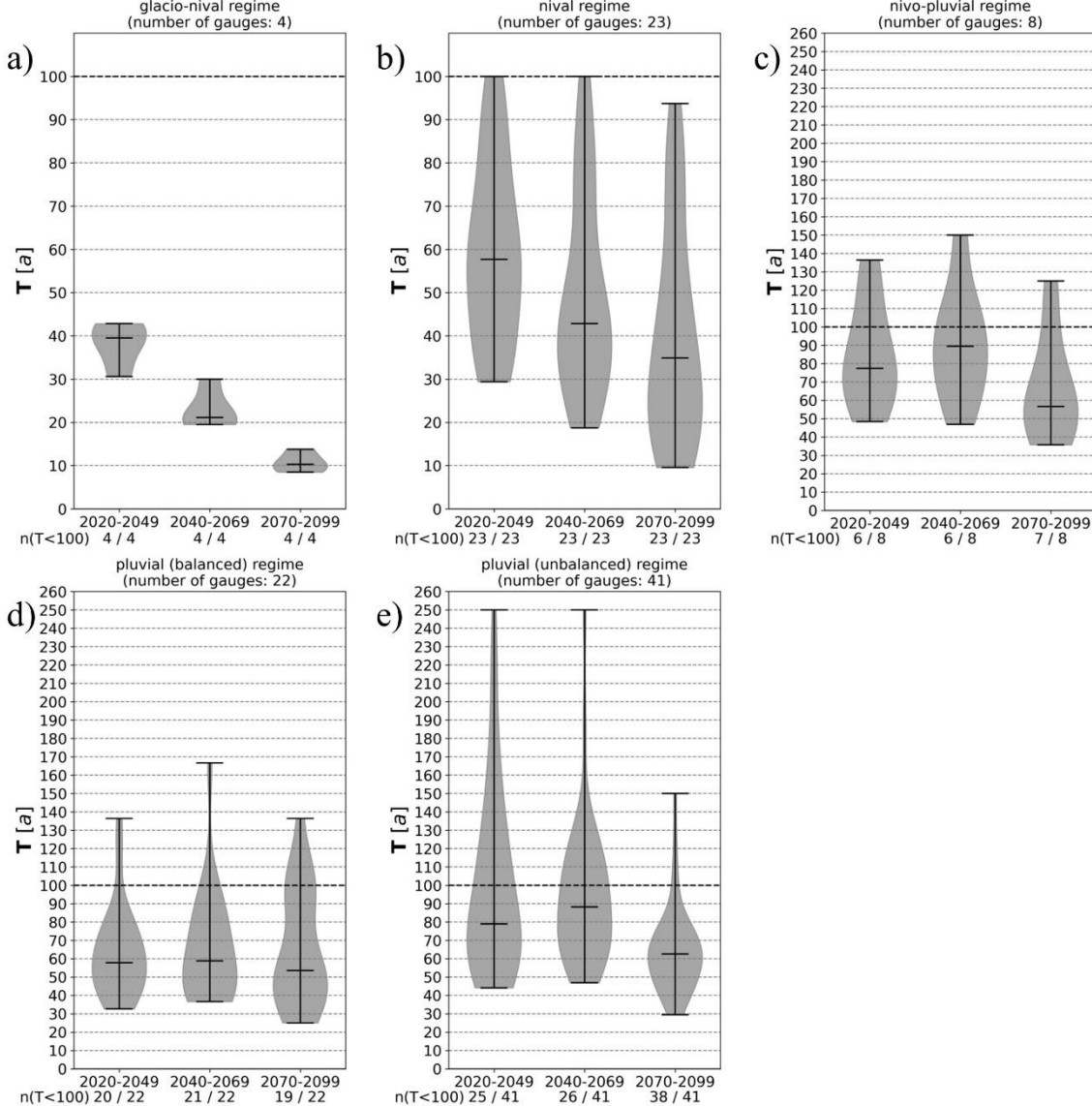

**Figure 6: Violin-plots indicating the changes of the frequency of the HF$_{100}$ for the three future periods (near, mid, far) compared to the reference period, with changes presented as absolute values of return periods (T[a]) of the respective future period compared to the 100-year return period for each gauge. Results of the 98 gauges are aggregated for the five discharge regime (a = glacio-nival, b = nival, c = nivo-pluvial, d = pluvial (balanced), e = pluvial (unbalanced)). The figures display the total number of gauges per regime as well as the number of gauges depicting an increased frequency.**

Values indicate the new return period associated with the HF$_{100}$ discharge from the reference period. This means values below 100 indicate an increase in frequency. The glacio-nival regime (Figure 6a) also exhibits the strongest increase in frequency among all regimes with the HF$_{100}$ of the past becoming equivalent to a 31- to 43-year event

in the near future, thus becoming roughly two to three times more frequent. For the 2085 horizon the same $HF_{100}$ event becomes an 8- to 14-year event showing a seven to twelve-fold increase in frequency. A similar development is visible for some gauges in the nival regime (Figure 6b). While the violin plot for this regime indicates that the reference 100-year event will become a 70-year event for more than 50% of gauges, some gauges show no or only a minor increase in frequency as well. The changes for the remaining regimes are less severe, but still indicate an increase in frequency for up to 50 % of the respective gauges until the middle of the century and more than 50 % in the far future. The changes for the nivo-pluvial regime (Figure 6c) and the unbalanced pluvial (Figure 6d) regime show that the frequency declines for less than 50 % of the gauges in the near and mid future period. Therefore, the 100-year event becomes more frequent for more than 50 % of the gauges with varying extent. While the magnitude of changes is similarly moderate (except for the far future) for Figure 6c and Figure 6e, projected future return periods for the $HF_{100}$ event for Figure 6d depict stronger change signals towards higher frequencies with more than 50 % of gauges showing values smaller than 60 years. Furthermore, the nivo-pluvial as well as the balanced and unbalanced pluvial regimes exhibit a slight decrease in frequency in the mid future compared to the remaining projection periods while the magnitude does not show this behavior. However, this circumstance may be explained by the change in driving agent from snowmelt driven events in the near future to rainfall induced events at the end of the century. Thus, at the 2055 horizon the shift of the ratio of both event types contributes to this slight decline in frequency.

Some gauges within the nivo-pluvial and both pluvial regimes depict an in parts large decrease in frequency and/or magnitude. These gauges usually exhibit natural or artificial influences, such as the retention effect of natural lakes, reservoirs, or diversions or gauges of small catchments which might experience less dynamics in changes of flood drivers or even a reduction.

Overall, the changes in frequency and magnitude due to the projected changes in climate according to the CRCM5-LE become less severe with increasing distance from the Alps. Furthermore, the increase in frequency and magnitude for alpine catchments is seemingly high, but in line with the results of Hattermann et al. (2018), which showed comparable results for the near future period (100-year event frequency between 20 and 40 years). The influencing factors for these in parts severe changes are manifold. However, Brunner et al. (2021b) analyzed the relation between the extremeness of precipitation and discharge for 78 out of the 98 gauges within Hydrological Bavaria and concluded that an increase in extreme precipitation magnitude is of higher importance for extreme return levels of discharge than land surface processes, such as antecedent soil moisture or changes in snowpack due to warmer temperatures. If precipitation volumes are sufficiently large, they quickly saturate the soil or yield an excessive amount of direct runoff due to infiltration excess (Brunner et al., 2021b).

The mean magnitude of the annual maximum precipitation is projected to change for different temporal aggregation levels (3-hourly to 5-daily) in the CRCM5-LE (Wood and Ludwig, 2020), as well as the magnitude of 100-year return period rainfall increases by 10-20% and the frequency increases by 2 to 4 times (Martel et al., 2020) for Hydrological Bavaria. The changes are associated with seasonal shifts from summer to winter events and are particularly pronounced in the Alpine region (Martel et al., 2020; Wood and Ludwig, 2020). Severe floods that occur simultaneously in different catchments of the study area are usually associated with a cutoff low Vb cyclone that results in prolonged precipitation events lasting up to 15 days over the same region (Stahl and Hofstätter, 2018; Mittermeier et al., 2019). Under changing climate conditions projected by the CRCM5-LE by the end of the 21$^{st}$ century employing the RCP8.5 scenario, these events are likely to intensify in volume and frequency during winter and spring and occur less frequently during the summer months but with an increased precipitation volume (Mittermeier et al., 2019).

The spatial distribution of the dynamics in $HF_{100}$ frequency and magnitude is shown in the supplements (S6).

### 3.3 Changes in driving mechanisms

Figures 7 shows the dynamics of three different combinations of driving mechanisms (columns of panels) listed in Table 1 for extreme discharge events equal to and above the $HF_{100}$ between the reference period (REF) and the three future periods (FUT1 to FUT3) for the five different discharge regime types (rows of panels). As mentioned in the methodology section, the 32 possible combinations were aggregated to six groups as listed in Table 2.

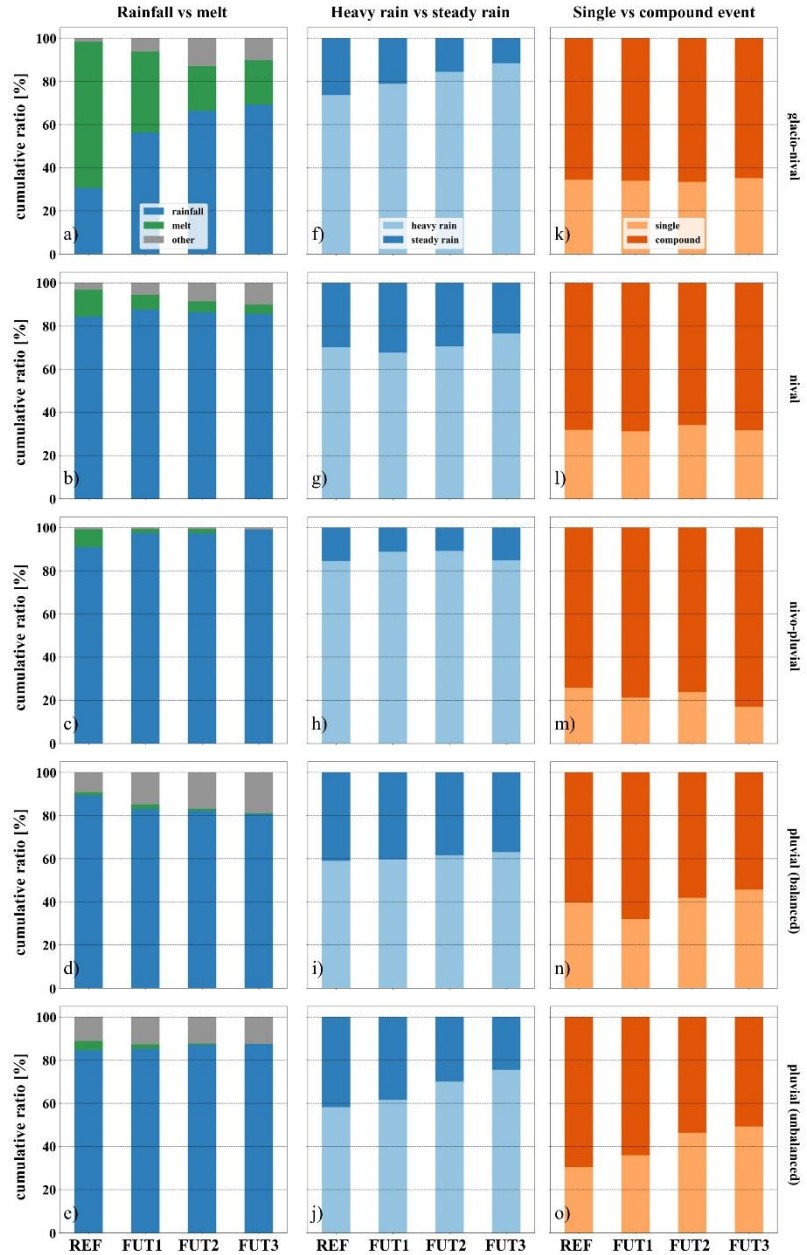

**Figure 7: Dynamics in driving mechanisms of floods equal to or larger than the HF$_{100}$ for the five different discharge regimes and its change over time. The columns of panels show the composition of melt vs. rainfall driven events (left), heavy rainfall vs. steady rainfall events (center) and single vs. compound events (right). The rows of panels correspond to the five different discharge regimes as indicated by the regime type on the right y-axis. The bars in each individual panel show the cumulative ration of the different mechanisms for the reference period (REF, 1981-2010) in the first bar and the following bars represent the different future periods (FUT1: 2021-2040; FUT2: 2041-2070; FUT3: 2070-2099).**

The first column of panels in Figure 7 show the changes in the ratio between snow melt driven events (excluding rainfall compounds) and rainfall driven events (including melt compounds). Panels f) - j) show the changes in the ratio between heavy rainfall and steady rain events (excluding snow melt events from Figure a) - e)), and panels k) - o) depict the changes in the ratio of events that can be attributed to a single cause and to a compound of drivers. The glacio-nival and the nival regimes show the highest ratio of snow melt driven events (Figure 7a and b). For

the reference period, this event type is the major driver for the extreme discharges for the glacio-nival regime (67.7%) while for all other regimes and periods rainfall and its compounds dominates the ratio. Figures 7a to e also show a 7th category 'other' which in this case comprises events which could not be ascribed to any of the investigated drivers and are likely events that originate from an upstream flood. All regimes indicate a decrease in snow melt driven event types form the REF period to the far future (FUT3) period, with the largest decrease visible

for the glacio-nival regime (67.7% to 20.6%). In the nivo-pluvial and pluvial regimes (Figures 7c to e) the ratio of snow melt event types becomes neglectable in the future with values of only slightly above 0%. Furthermore, the ratio of snow melt events diminishes from the glacio-nival and nival regime to the pluvial regimes (Figure 7c to e) with an increasing distance to the Alps. The pluvial regimes (Figure 7d and e) show only minor changes in the ratio of rainfall events from REF to FUT3 (89.3% to 80.6% for the unbalanced and 84.6% to 87.5% for the balanced

regime). With snow melt driven events disappearing in future periods, this indicates that more events are driven factors falling in the category 'other' with an increase of this category especially in the nival (3.2% to 10.1%) and unbalanced pluvial regime (9.2% to 18.9%). Overall, these results indicate a reduction in snow accumulation during the winter due to an increase in winterly temperatures between 3°C and 5°C as projected by the driving CRCM5-LE for the end of the century (FUT3) for middle Europe (von Trentini et al., 2020). For the glacio-nival

and nival regime the large decay of snow melt ratio also indicates a reduced contribution of melt from glaciers due to severe loss of mass towards the far future.

    Figure 7 f) – j) illustrate the dynamics in the ratio of hr and sr event types, therefore representing only the rainfall part of Figures 7a to e. In all five regime types, the hr event type and its compounds (including compounds with sr) are the dominant driver with a ratio of 58.8% in the pluvial unbalanced regime (REF) to 88.4% in the glactio-

nival regime (FUT3). The percentage of hr events increases towards the end of the century for all regimes by 4 (pluvial balanced) to 17.2 (pluvial unbalanced) percent points except for the pluvio-nival regime, where the percentages first increase from REF in FUT1 (88.9%) and FUT2 (89.2%) and then decrease again for FUT3 to the level of REF (REF: 84.4%; FUT3: 84.8%). The glacio-nival and pluvial unbalanced regimes show the strongest increase in hr event types from REF to FUT3 with 14.7 and 17.2 percent points respectively. The dynamics for

these regimes may be caused by an increase in summerly (between 5°C and 6.5°C), but also spring and fall temperatures towards the end of the century as projected by the CRCM5-LE (von Trentini et al., 2020). The higher

temperatures result in more available water vapor and thus, precipitable water in the atmosphere and a higher potential for convective (hr) events, especially over the Alps (Giorgi et al., 2016). Furthermore, the strong increase for the alpine gauges of the glacio-nival regime may be related to a stronger increase of heavy precipitation events over the Alps compared to regimes outside the Alps (Wood and Ludwig, 2020). In general, the balanced and unbalanced pluvial regimes show the lowest contribution of hr events (59% and 58.3% in REF) and therefore any change in the number of hr events or a general increase in the intensity of short duration rainfall might lead to more events being classified as hr compounds. Whereas in other regimes the hr compound is already large and hence, any changes in the rainfall dynamics will only yield a limited increase in the event classification.

Figures 7 k) – o) illustrate the dynamics in the ratio between single driver event types and compound event types for the five different regimes. The compound class here complrises the snow melt and rainfall event types in Figures 7 a) – e), neglecting events classified as 'other'. Therefore, single driver event types depict events caused by only one of the driving mechanisms listed in Table 1, whereas compound event types comprise all other possible combinations. In all five regims and time periods, compound drivers are attributed to at least 50.7% (FUT3, pluvial unbalanced) up to 82.9% (FUT3, nivo-pluvial) of events. Except for the two pluvial regimes (balanced and unbalanced) there is only very little change in the ratio. For these regimes the number of events caused by a single driver increases form REF to FUT3 by 6 and 18.8 percent points for the balanced and unbalanced regime respectively. This strong signal in the dynamics for the unbalanced pluvial regime indicates an increase in short events of high intensity which in turn may lead to a higher risk for flash floods. The nivo-pluvial regime further depicts a slight decrease in events caused by a single driver from REF to FUT3 by 8.8 percent points.

**4 Discussion**

The variability of statistical characteristics within a time series can affect the estimation of extreme values due to extraordinary events (Fischer and Schumann, 2016). The results of this study emphasize the benefit of using data provided by a climatological SMILE for hydrological impact studies as it provides a profound basis for extreme value statistics and allows for more accurate estimation of extreme values, as also shown by other studies (van der Wiel et al., 2019; Champagne et al., 2020; Ehmele et al., 2020; Maher et al., 2021). In particular, van der Wiel et al. (2019) follow a similar approach for comparing statistical distributions and an empirical approach to derive extreme discharge values by employing a different approach for the creation of a 2000-year large ensemble of climate data forcing a global hydrological model. Our study shows similar results for the comparison between statistical estimates and empirically derived $HF_{100}$ values favoring the empirical over the statistical approach when

using a large ensemble of hydrological model data. However, the in parts large deviations between the benchmark (robust estimate derived from the empirical probability for a 100-year flood event using 1,500 AM values) and the estimates derived using a GEV based on different sample sizes (30, 100, 200) might be reduced when using an extreme value distribution which is better suited for the respective sample when enough data is available (i.e., when using a hydro-SMILE as shown here). In some cases the GEV might not be the best distribution for the samples of the respective gauge which might affect the differences from the benchmark since higher quantiles heavily depend on the distribution (Schulz and Bernhardt, 2016). Hence, a large ensemble of hydrological model data further reduces the uncertainties originating from different distribution which may be considerable as shown by Lawrence (2020). However, the approach presented in this study illustrates the benefit of a hydro-SMILE as it provides a more robust estimate by employing empirical probabilities for the deduction of extreme values. Therefore, these robust estimates allow for a more robust assessment of future dynamics of extreme high flows as well as their driving mechanisms as also shown in Brunner et al. (2021b).

The results of this study are subject to uncertainties (parameter, process description) as they are produced by data created at the end of a cascade of modeling steps usually applied for climate change impact studies as displayed in Figure 2. Different components (e.g., climate model, hydrological model, bias correction) affect different discharge characteristics or indicators (e.g., extreme indicators, mean discharge) (Gampe et al., 2019; Muerth et al., 2012; Muerth et al., 2013; Velázquez et al., 2013; Willkofer et al., 2018). A thorough assessment of the contribution of the chain compartments to the overall uncertainty would require an ensemble of multiple climate and hydrological models.

The overall strong increase in frequency and magnitude of the $HF_{100}$ in the future may be driven by deficiencies of the employed hydrological model, such as generalized glacier model among affected catchments, or a single snow melt approach for the entire Hydrological Bavaria (as described in Willkofer et al., 2020). However, as stated in the previous section, this scale of change was also found by Hattermann et al. (2018) for the upper Danube basin using the same emission scenario projections, but a different hydrological and climate model, which might indicate that the change signals are likely independent of the chosen hydrological or climate model. However, several gauges north of the Alps exhibit a decrease in frequency and magnitude of $HF_{100}$ over the three different future periods compared to the reference period. As mentioned, these gauges are in parts affected by artificial or natural retention (e.g., reservoirs) or transfer systems which are implemented in the model and thus may influence the results. Additionally, despite the projected increase in extreme rainfall events of the CRCM5-LE even north of the Alps (Wood and Ludwig, 2020), the non-linear behavior of the processes involved in runoff generation may not translate this increase into extreme discharge events (Brunner et al., 2021a). Furthermore, this increase in extreme

rainfall events is less severe north of the Alps (Wood and Ludwig, 2020), which may further contribute to the decline or minor increase in frequency and magnitude of the $HF_{100}$ events .

The results of the CCI on the frequency and magnitude also depend on the performance of the hydrological model. Since it relies on observations for parameter calibration, the quality of this data is crucial, especially for extreme values. For the most extreme events (e.g., $HF_{100}$ and above) the river may inundate the surrounding area and the water level / discharge relationship at the gauging station used to determine discharge values may not be valid anymore and is likely to underestimate the peak discharge. Therefore, the actual observed discharge – and thus,

the calibrated model – is prone to these measurement uncertainties. This is a general limitation in hydrological modeling. Furthermore, the discharge of rivers within Hydrological Bavaria is heavily impacted by management structures for flood protection or hydro power generation, especially the southern tributaries of the Danube in the Alpine foreland and within the Alps are heavily regulated. Since the management follows somewhat fuzzy rules and actual data is restricted by private companies in most cases, the management rules for these structures must

be deduced from publicly available data and implemented in the hydrological model. These rules are susceptible to extreme conditions as they do not allow for adaptations during model runtime (e.g., flushing a reservoir prior to an anticipated heavy precipitation event).

The projected future changes in extreme discharges may be attributed in part, to the climatological reference dataset, as it affects the performance of the hydrological model as well as the CCS through bias adjustment (Gampe

et al., 2019; Meyer et al., 2019; Willkofer et al., 2018). Precipitation in high altitudes (e.g., the Alps) may be under-captured (Westra et al., 2014; Poschlod, 2021; Prein and Gobiet, 2017; Rauthe et al., 2013; Poschlod et al., 2020; Willkofer et al., 2020) resulting in an underestimation of observed precipitation in these regions, especially of extreme values. Assuming a temporally stationary bias, changes in the extremes might be overestimated due to an over-adjustment of the distribution of the reference period towards underestimated observations compared to the

future periods. Furthermore, the variables are adjusted individually and thus, physical coherency as for a multivariate approach proposed by Meyer et al. (2019) is not guaranteed. This specifically affects discharges governed by snow or glacier melt of higher elevation within the Alps (Meyer et al., 2019).

The analysis of the dynamics in driving mechanisms of extreme discharges of the $HF_{100}$ and above involved a set of thresholds for several parameters (rainfall, snow/glacier melt, soil water content) as well as their combinations.

Thresholds other than those selected according to the DWD (Deutscher Wetterdienst, 2024) may yield different ratios of the illustrated dynamics. The different drivers and their different combinations considered for this analysis could have been aggregated to other overarching categories (e.g., showing the contribution of soil moisture). However, we opted for the illustrated aggregations as changes in the extremeness of these events directly translate into changes in extremeness of the discharge (Brunner et al., 2021a). Furthermore, the analysis only focuses on

discharge events which are above the benchmark 100-year flood event calculated for the reference period. Hence, for gauges depicting a decrease in $HF_{100}$ frequency and magnitude in future periods events resulting from the changes in future return levels are not considered here. However, it is unlikely that the overall dynamics gained from this approach might considerably change by applying the future $HF_{100}$ values as threshold to extract the events.

Since the presented modelling approach only comprises one GCM-RCM combination forced by the more extreme RCP8.5 emission scenario as well as one hydrological model, the significance of the findings regarding the variance of change effects in the future on the development of extreme peak flows is limited. Furthermore, the projected climate change signals of the CRCM5-LE were found to depict a stronger warming and drying compared to other large ensembles (von Trentini et al., 2020) which might result in these part extreme increase in frequency

and magnitude of the $HF_{100}$ values among many gauges of Hydrological Bavaria.

Projected discharge extremes at the upper end of the distribution that have not been observed to date might be created by unrealistic compound events due to flaws in the bias correction approach (Kelder et al., 2022). Thus, these events directly influence the EVD, producing higher return values, and consequently, a larger change signal. However, as extreme precipitation events of various durations are expected to intensify within the studied region,

the probability for yet unseen floods due to compounding events may also increase in the future.

**5 Conclusion**

This study emphasizes the benefit of employing a climatological SMILE with a hydrological model to create a hydro-SMILE to foster extreme value statistics and analyze the impacts of climate change on hydrological extreme values such as the $HF_{100}$ due to the provision of a very large database. This database allows for the application of

empirical exceedance probabilities to estimate robust discharge values of high return periods rather than statistical extrapolation based on extreme value distributions. The results show that the performance of statistical estimates largely depends on the available length of the time series as well as its values when compared to the empirical benchmark. However, even with a length of 200 AM, the variance of the scatter of $HF_{100}$ estimates of the 1,000 samples was rather large.

As mentioned by Willkofer et al. (2020) the performance of the hydrological model allows for CCI studies - in this case using the CRCM5-LE to elaborate on the effects of climate change on the development of the $HF_{100}$. The projections reveal a strong increase in the magnitude and frequency of $HF_{100}$ events for Alpine and pre-Alpine catchments exhibiting a snowmelt driven discharge regime within the reference period. This strong increase in the magnitude and frequency is considerably smaller for catchments north of the Alps and of a more pluvial discharge

regime. The in parts tremendous changes of $HF_{100}$ intensities and frequencies may be ascribed to the emission scenario (RCP8.5). Thus, the addition of different SMILEs and hydrological models may foster the significance of the findings due to different climate projections and simulated climatological and hydrological processes along the model chain. However, the establishment of such extensive model chains requires vast computational resources. Nevertheless, this effort should be considered regarding the benefits this profound database offers for

extreme value statistics, fostering the knowledge about the propagation of natural variability of the climate system to the hydrological response (Brunner et al., 2021b), or allowing to distinguish climate change signals (or forced response) from natural variability for extreme values (Wood and Ludwig, 2020; Aalbers et al., 2018).

This study further shows the benefit of a hydro-SMILE when driven by a processed based hydrological model as it allows for a more detailed analysis on the processes responsible for the genesis of such extreme discharges.

Within the study area extreme discharges events larger than the $HF_{100}$ are less likely to be caused by snow melt events in the future as higher winterly temperatures will result in less snow accumulation. Hence, rainfall becomes the dominant driver in the future. Further, those events are more likely to be caused by heavy than steady rainfall in the future, although the degree in dynamics may vary for the different regimes. While compound events of superimposing drivers might remain the major cause for discharges equal to and greater than the 100-year flood,

the number of events caused by a single driver such as heavy rainfall is likely to increase in the future at least for the two pluvial regimes.

Furthermore, the results highlight the need to incorporate climate projections in the design of new flood protection infrastructure or adapting existing structures to reduce future flood risk, not only in Hydrological Bavaria, but everywhere in general. Further studies are necessary focusing on flood inundation to fully analyze the extent of

the increase and frequency of this event for the design of flood protection infrastructure.

**Contributions**

FW developed the concept of this study including methods, investigation, and visualization, performed the formal analysis and validation, and prepared the original draft. FW and RRW were responsible for data curation and software development. RL acquired the funding for the presented research, provided the required resources, was

responsible for the project administration, and supervised the presented research.

**Competing interests:**

The authors declare that they have no conflict of interest.

**Acknowledgements**:

The authors acknowledge the colleagues Gilbert Brietzke, André Kurzmann, and Jens Weismüller from Leibniz
Supercomputing Centre of the Bavarian Academy of Sciences and Humanities for their technical support during
the development of the hydrological large ensemble. We also thank the Leibniz Supercomputing Centre for
providing the HPC infrastructure and computation time.

F.W. was supported by the ClimEx project, funded by the Bavarian Ministry for the Environment and Consumer
Protection.

The CRCM5-LE was created within the ClimEx Project, which was funded by the Bavarian Ministry for the
Environment and Consumer Protection. Computations of the CRCM5-LE were performed on the SuperMUC HPC
system of the Leibniz Supercomputing Centre of the Bavarian Academy of Sciences and Humanities. We
acknowledge Environment and Climate Change Canada for providing the CanESM2-LE driving data.

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
