# Peer review of "Assessing the impact of climate change on high return levels of peak flows in Bavaria applying the CRCM5 Large Ensemble"

_EGUsphere, 2023_

## Author Comment (AC1)

Author's response to the comments of reviewer 2:

Summary: The manuscript employed a large ensemble of hydrologic simulations driven by a climate model (Hydro-SMILE) to investigate the future changes in peak flow characteristics and dynamics. Hydro-SMILE can be a powerful tool for extreme value estimates in hydrology. The approach, based on validated physics-based hydrologic models and large ensemble climate data, is especially helpful for understanding the changes in magnitude, frequency, and dynamics in high-return level peak flow events. However, the authors did not fully utilize the data they generated and the focus of the manuscript has been on how beneficial such a tool can be. Further and more in-depth analyses on the changes in the characteristics of high return level peak flow events are needed to improve the manuscript. See below my major comments. In addition, I provide language edits/suggestions in minor comments for places where I think clarity is lacking.

AC: The authors would like to thank the reviewer for the valuable comments to further improve the manuscript. Detailed author comments (AC) on the reviewer comments (RC) are provided for each individual comment.

Major comments:

1. The manuscript has a title of "…high return levels of peak flows…" but specifically focused on 100-year floods. With a large ensemble, the authors could investigate a range of high return levels of peak flows and see how the frequency, magnitude, and dynamics are projected to change in the future climate.

AC: Thank you for your comment. You are right, the large ensemble would allow for the analysis beyond the 100-year return period, however, despite the extensive data available for analysis, uncertainties calculating return periods beyond e.g., 300-year floods would increase again and the sample size for analyzing the drivers is limited. Further, to be consistent with the storyline of the manuscript and the relevance of the 100-year flood in the study region (i.e., design criteria for flood protection or hydropower infrastructure) we prefer to remain the focus on HF100. We will however extend the analysis of the 100-yr Flood and the climate change impact on the dynamics and flood generating processes, as recommended in your review comment 4. With this extension of the analysis showing results for multiple return periods for all these different topics would lead to complex figures and excessive descriptions which may be repetitive in many places.

2. The manuscript focused on proving that it is beneficial to have a large ensemble to estimate extreme peak flow events (Sections 3.1 and Figures 4 and 5), which, I think, is very obvious so I suggest moving them to the supplementary. The authors have generated a powerful dataset for extreme peak flow estimation, however, there is no analysis of the changes in flood frequency and magnitude. The authors are suggested to substantially expand the analysis on flood frequency and characteristics. See Yu et al. (2020) for an example of flood frequency analysis.

Yu, G., Wright, D. B., & Li, Z. (2020). The upper tail of precipitation in convection-permitting regional climate models and their utility in nonstationary rainfall and flood frequency analysis. *Earth's Future*, 8,

e2020EF001613. https://doi.org/10.1029/2020EF001613

AC: The authors partly agree with the comment on the obviousness of the benefit of large ensemble. We think that this is true for some research areas, such as climate/atmospheric sciences where they can be considered a state-of-the-art tool. However, in hydrology large ensembles are still rarely used and rather statistical methods (e.g., weather generators, emulators) are more frequently used to artificially enhance the sample size. We however think that the use of large ensembles in hydrological studies brings many advantages over these statistical methods, especially when it comes to changes in the dynamics. Thus, we think that highlighting the benefits of the represented approach is still necessary and valuable to make these tools more known and accessible within the hydrological community.

Regarding your comment on "…there is no analysis of changes in flood frequency and magnitude." We disagree and point to figures 6 and 7 depicting the changes in frequency and magnitude (intensity), although presented differently compared to your mentioned reference Yu et al., 2020. Yet, the authors will consider the suggestion to further elaborate on the changes in characteristics (as further mentioned in comment 4) and incorporate additional figures and explanations. For this we further consider shortening the part on the benefits (e.g., move Fig. 5 to the supplement materials).

3. The authors only gave limited information on the evaluation results of the hydrologic model which are essential for building confidence in the following analysis. I suggest including figures and/or tables showing the evaluation results such as time series of flow events with other quantitative metrics such as correlation coefficient, % bias, root mean square error, KGE, NSE, etc. One related question would be how the level of trust (LOT) is calculated.

AC: We will add additional information on the model's performance in the supplement materials. Furthermore, the authors will refer the reader to specific publications for additional information about the model performance, as the mentioned metrics and figures have already been published in several papers. We will likely keep this short in the main manuscript and will include some figures in the supplementary material.

4. The changing dynamics in the future climate (Section 3.2) are very interesting and worth digging into. The authors could dig into it with a mechanistic investigation of possible explanations for why they see such changes in future projections. For example, linking snow water equivalent and rain characteristics to the dynamical changes that are projected at the nivo-pluvial stations.

AC: We agree and will provide further analysis and figures showing the changes in flood driving mechanisms which may explain the depicted changes in flood dynamics for the presented flow regimes.

Minor comments:

Lines 57-59: it is unclear how prediction is a reason for challenges in modeling and predicting high flows by Brunner et al. (2021a).

AC: The Authors will remove prediction from the reasons for challenges in modelling and predicting high flows, as this was wrongfully stated by the authors.

Line 71: change "extraordinary" to "extreme".

AC: The authors will exchange the term as recommended.

The paragraph starting from line 73: Needs substantial rephrasing. 1) Rephrase the sentence "This approach of high spatiotemporal resolution for climate and hydrological modeling is computationally demanding." Do the authors mean "This ensemble-based climate and hydrological modeling approach is computationally demanding because of the high spatio-temporal resolution"?

AC: The authors will rephrase the mentioned paragraph for better clarity.

2) Rephrase sentence "However, considering spatially refined catchment features (e.g., slopes, soil characteristics, land use), precise values due to higher temporal resolution, and the application of a SMILE for hydrological modeling supports an enhanced representation of extreme values within models." Do the authors mean high spatio-temporal resolution of hydro-SMILE is particularly valuable for an enhanced representation of extreme values in models because hydro-SMILE considers spatially-refined catchment features at high temporal resolutions?

AC: The reviewer's interpretation of the sentence is correct; the authors will rephrase it accordingly.

3) Rephrase "Thus, this study focuses on the major Bavarian river basins (upper Danube, Main, Inn) with all their tributaries". It is unclear to me what your study area has to do with the above two statements.

AC: The authors will rephrase this sentence to better explain that a high spatio-temporal resolution is beneficial and necessary for the heterogeneity within the study area.

Line 84: Remove "Therefore".

AC: The authors will remove it.

The paragraph that starts from Line 84: add section numbers throughout the paragraph.

AC: The authors will add section numbers as recommended.

Line 85: Confusing sentence. Remove "…to meet the requirements for the hydrological modeling."

AC: The authors will remove this part of the sentence in Line 85.

Line 86: "…hydro-SMILE along..." should be "…along with…".

AC: The authors will change this phrase according to the suggestion.

Line 95: Remove "As a result"

AC: The authors will remove this phrase.

Line 102: "…(up to 1100 mm precipitation sums in the north, 2500 mm in the south; an average temperature of 10 °C in the north, down to 5 °C (-8 °C on alpine summits)…". Are

the authors referring to annual total precipitation and annual mean temperature? Be clear on that. Also, be sure to mention the data sources for these numbers - are they from Poschlod et al. (2020) as well?

AC: The authors will clarify on the actual meaning of the mentioned precipitation sums and average temperatures and will further add references for these numbers as well.

Line 108: "The major river catchments were divided into a total of 98 smaller sub-catchments based on a common interest in flood protection and a more detailed variation in catchment characteristics, using a selection of gauges (Willkofer et al., 2020)." Rephrase this sentence. It is unclear to me whether the 98 sub-catchments were divided based on the spatial distribution of the 98 selected gauges or whether the gauges were selected because of the division of the 98 sub-catchments.

AC: The authors will rephrase the sentence for a better understanding of why the mentioned 98 catchments were selected for this study.

Line 125: Figure caption of Figure 2: Also introduce what are SDCLIREF and WaSiM in the Figure caption.

AC: The authors will add the explanation for SDCLIREF and WaSiM to the Figure caption.

Line 142: What is "T63"?

AC: T63 is a term describing the original grid resolution of the climate model. As it does not contribute to further understanding of the data, the authors will simply remove this term.

Line 149: "Furthermore, the individual members of the CRCM5-LE are considered independent for the hydrological evaluation period from 1981 to 2099, as the analysis of variations in temperature and precipitation over land and ocean shows (Leduc et al., 2019)." This is the first time the authors mention "hydrological evaluation period". This is a confusing sentence. Aren't the CRCM5-LE individual members independent no matter what time period?

AC: We will rephrase this sentence and likely move this sentence to be clearer. Please also see our response to your comment for Line 214 below. For the period that is considered in this paper (1981-2099) the individual members are indeed fully independent of each other. Only at the start of the simulation (1950-) there is a spin-up period which is needed for the members to achieve the full spread of variability and hence to become independent of each other. This spin-up period is inherent from the chosen initialization scheme (macro and micro). The micro initialization (which are introduced by a very small rounding error) all start from the same ocean conditions (within the five different parent simulations) and it takes the system a few years (up to five years) to develop different ocean states which then makes the members fully independent of each other. This spin-up behavior is shown by Leduc et al (2019) for the respective climate simulations used here. We will rephrase this paragraph to clarify this.

Line 152: "…showing regional and seasonal variations in magnitude over Europe (Leduc et al., 2019)." What variables do the authors mean?

AC: The authors will add the variable (temperature and precipitation).

Line 156: add "match" after "were adjusted to".

AC: We will rephrase.

Line 157: Change "RCM scale" to "RCM grid". Did the authors do the interpolation onto the RCM grid? If yes, be clear on what interpolation scheme is used.

AC: The authors will change "RCM scale" to "RCM grid" and we will include the interpolation scheme used to upscale the meteorological reference dataset to the RCM grid. We used a mass-conserving approach.

Line 160: "…3-hourly correction factors for every quantile and month". Unclear how the 3-hourly correction factor is applied.

AC: We will clarify this. Correction factors were determined for each quantile bin for each month and sub-daily (3h) time step. To preserve the ensemble spread, all members were pooled to obtain the correction factors and these factors were subsequently applied to each ensemble member separately.

Sentence starting on Line 162: I think the authors want to stress that bias correction is inevitable. Rephrase it to "Despite the benefits (increasing reliability of climate change projections of the hydrological impact model, reducing bias in mean annual discharge) and shortcomings (disrupting feedbacks between fluxes, modification of change signals, assumption of a stationary bias) of bias correction are highly debatable (e.g., Teutschbein and Seibert, 2012; Maraun, 2016; Ehret et al., 2012; Dettinger et al., 2004; Chen et al., 2021; Huang et al., 2014), bias correction is often inevitable for climate change impact studies (Gampe et al., 2019)."

AC: Thank you for this suggestion, we will rephrase the sentence accordingly.

Line 168: For such a topographically complex region as described in the "Study area" section, I'm concerned the statistical downscaling between grids that are so different (from 12 km in RCMs to 500 m in hydrologic models) will lose important spatial heterogeneity across the domain. Does the mass-preserving approach address this problem?

AC: The mass-preserving approach ensures that through the downscaling no additional precipitation is added or lost. Meaning, when the downscaled result is upscaled to the RCM scale the mass of the RCM grid is matched. The downscaling basically tries to distribute the coarse RCM information to a higher sub-grid scale. We don't think that the downscaling will alter the spatial heterogeneity. If so, then rather the bias correction will alter the spatial heterogeneity. As with all interpolation schemes, the obtained spatial result is only one of many possible spatial distributions.

Line 171: "The interpolation result was then applied to the SDCLIREF reference fields (Brunner et al., 2021b)". Unclear. Do the authors mean the SDCLIREF reference fields are also interpolated using the same method?

AC: No, the SDCLIREF reference fields were generated by combining a multiple-linear regression (considering, elevation, slope, longitude, and latitude) and inverse-distance weighting. The interpolation method is based on the interpolation method used by the German

Weather Service (DWD, Rauthe et al. 2013). We will add the necessary information here to clarify this.

Line 194: "minimizing a weighted combination of performance metrics, including Nash and Sutcliff efficiency (NSE; Nash 195 and Sutcliffe, 1970), Kling-Gupta efficiency (KGE; Gupta et al., 2009), the logarithmic NSE and the ratio of root mean squared error to standard deviation (RSR; Moriasi et al. (2007)) (Eq. (1))". Introducing the overall metric (OM) equation first and then describe what is in the equation. Then, give a threshold - what OM value is considered "good" or "bad"?

AC: The authors will introduce the overall metric first and then describe it. However, the DDS-SA approach will always try to minimize this overall metric regardless of the threshold (although one could introduce a threshold to reduce the number of iterations once the threshold is reached). Once it reaches 0 the algorithm stops. However, provided that an unlimited number of iterations is applied, the algorithm will go on trying to minimize the metric's result. The authors did not add an additional threshold which terminates the algorithm once this threshold is reached. Thus, we could provide a threshold where we consider the model to be good or bad, but in terms of evaluation this threshold would not be of much significance.

Line 203: "(NSE: 16; KGE: 5)" Unclear. What does this mean?

AC: These numbers mean, that 16 (5) gauges out of 98 did not perform sufficiently well (lower than 0.5) for NSE (KGE). We will clarify this.

Line 208: "Consequently, the level of trust (LOT) for peak flows of return periods of 5, 10, and 20 years of flood events, introduced in Willkofer et al. (2020) showed a moderate to high confidence for most catchments, with gauges of poor simulated performance yielding a lower LOT with increasing return levels." Do the authors mean gauges with good performance have higher LOT for peak flows with return periods of 5, 10, and 20 years, whereas gauges with poor performance have lower LOT, especially for peak flows at longer return periods?

AC: The reviewer is correct. The meaning of this phrase is misleading, and the authors will thus rephrase this sentence to clarify the meaning of the LOT in this case.

Line 214: "The entire modeling period is shortened by ten years to account for the time span it takes the RCM to produce fully independent realizations due to the inertia of the ocean model (Leduc et al., 2019)." I have two comments: 1) I saw that the authors partially address my question for Line 149 here. It would be better to rearrange this part and the sentence on Line 149 such that the 10-year spin-up period and the choice of the evaluation time period are more clearly lined up and explained. 2) Rephrase this sentence to "We focus on 1961—2099 as opposed to 1950 – 2099 to account for the time it takes for the RCM to produce fully independent realizations due to the inertia of the ocean model (Leduc et al., 2019)."

AC: The authors will rearrange the paragraphs and will combine. Please also see our comment to line 149 above since they are connected.

Figure 3: What is "HF T,BM" on the far right?

AC: This term describes the benchmark (BM) high flow of return period T which is calculated using the empirical probability (p) and the entire series of 1500 yearly peak flows. The authors will describe this value in the Figure caption.

Line 256 and thereafter: Change "intensity" to magnitude throughout the manuscript.

AC: The authors will change the term throughout the manuscript.

Line 293: "…as indicated by the spread of the blue markers around the black benchmark line." I think the authors mean "… as indicated by the decreasing spread of the blue markers around the benchmark line with increasing sample size."

AC: The reviewer is correct. The authors will change this phrase accordingly.

Lines 296-297: What are panels a, c, and e?

AC: This is a clear mistake in referencing the different parts of the Figure. The authors will change the references to Fig. 4a, 4b, and 4c.

Figures 1&6: What do "balanced" and "unbalanced" pluvial mean?

AC: According to Poschlod et al. 2020, the term balanced describes a relatively even flow regime with only little variation in monthly mean flow and only little snowmelt influence during November to March. Poschlod et al. 2020 further describes the unbalanced pluvial regime as having a more pronounced peak in mean monthly discharge from January to March and a more pronounced decline during the second half of the year. The authors will add this information to the manuscript.

References:

Poschlod, B., Willkofer, F., and Ludwig, R.: Impact of Climate Change on the Hydrological Regimes in Bavaria, Water, 12, 1599, doi:10.3390/w12061599, 2020.

Willkofer, F., Wood, R. R., Trentini, F. von, Weismüller, J., Poschlod, B., and Ludwig, R.: A Holistic Modelling Approach for the Estimation of Return Levels of Peak Flows in Bavaria, Water, 12, 2349, doi:10.3390/w12092349, 2020

Brunner, M. I., Swain, D. L., Wood, R. R., Willkofer, F., Done, J. M., Gilleland, E., and Ludwig, R.: An extremeness threshold determines the regional response of floods to changes in rainfall extremes, Commun Earth Environ, 2, doi:10.1038/s43247-021-00248-x, 2021

---

## Author Comment (AC2)

Author's response to the comments of reviewer 1:

In this study, the authors assess the impacts of climate change on the 100-year flood events in several rivers in Bavaria region, in which most of the flood protection infrastructures are designed based on the 100-year flood events. Specifically, the authors explore a large ensemble of 50 members of CanESM2 climate model, which are produced using different initial conditions but the same model structure, parameters, and a high emission scenario (RCP8.5). A hydrological model is then used to simulate river discharge, followed by a dataset of 1500 model years (30 years of the reference period x 50 members), which are used for extreme value analysis. Also, the authors show the benefit of using hydro-SMILE in providing a more robust extreme hydrological discharge values under climate change impacts. In overall the study is beneficial for the study region. However, there are still some concerns, which require further improvements. My detailed comments are below.

AC: The authors would like to thank the reviewer for the valuable comments to further improve the manuscript. Detailed author comments (AC) on the reviewer comments (RC) are provided for each individual comment.

1. The fourth paragraph in Introduction (Lines 73-77, Page 3) is vague. The two first sentences do not seem to be the reason for the choice of study area.

AC: The authors agree with the comment and will rephrase the mentioned sentences to better point out the choice of this study area.

2. Can the authors explain the choices of CanESM2 and the use of RCP8.5 only?

AC: The CanESM2 large ensemble was chosen because of its availability of high temporal resolution outputs necessary for driving the CRCM5. Other CMIP5-era large ensembles have either not archived the necessary fields needed for dynamical downscaling, or it was too difficult to retrieve the data. Secondly, there was basically no choice of different RCP scenarios, since we were dependent on the driving GCM data. The CMIP5-era large ensembles were all ensembles of opportunity where mainly only one scenario was possible to run. Hence, this era of large ensembles has almost exclusively produced RCP8.5 simulations. In our project we also only had the opportunity to downscale one large ensemble.

3. The study area has abundant in situ data. Should model parameters (i.e., soil properties) be calibrated?

AC: The authors agree that the study area offers comparably very good in situ data. Yet, these data are mainly available as point measurement or spatially interpolated which means that there are inherent uncertainties. Hence, the authors consider a model calibration as necessary for the selected model. Furthermore, as the model is physically based, thus, not considered entirely physical, it allows for many free parameters to be set in the various modules which not all of them are measurable. The holistic modelling approach chosen for the hydrological model which provides a generalized parameter set for the entire region as described in Willkofer et al. 2020, aimed towards a reduction of these free parameters, thus reducing parameters to be calibrated. Furthermore, soil parameters were not calibrated (except for karstic areas where it was necessary to simulate faster infiltration). However, this approach had direct impact on the model performance as catchment specific characteristics were largely disregarded. The authors can briefly elaborate on this comment within the manuscript.

4. Since the reliability of the hydrological model affects the simulated discharge and the further analysis, the performance of the hydrological model should be presented and discussed in more detail. For example, for 16 gauges having NSE lower than 0.5 and 5 gauges having KGE lower than 0.5 (Lines 203-204, Page 7), an explanation is needed to show that the unsatisfactory is acceptable. For the other gauges with NSE and KGE higher than 0.5, how much higher than 0.5 are they? I think it is worth having maps that show the value of model performance metrics at all gauges.

AC: Maps of the mentioned metrics are already published in several publications (see: Poschlod et al. 2020, Willkofer et al. 2020). Detailed discussions on the matter are also presented in Willkofer et al. 2020 and Brunner et al. 2021. The authors therefore would refer the reader to these references for further reading on the model performance. We will briefly elaborate on the analysis within the manuscript without going into exhaustive details. In addition, we can provide maps of the performance metrics within the supplement material. Appropriate references to these figures will be placed in the main manuscript. In general, the gauges of the hydrological Bavaria are in parts heavily impacted by transfer systems, hydropower dams, flood protection measures, channels, as well as by local natural characteristics (e.g., Karst, shallow catchments with little variance in mean monthly discharge) which affects model performance due to the holistic approach.

5. Panels c, d, and e in Fig. 7 do not show the entire variation ranges.

AC: the authors agree and will change the scale of the figures to display the entire variation ranges.

6. I think it is worth having a map that visualizes the spatial variation of the change in return period under climate change impacts, some interesting insights might be found. I am curious about the difference between the changes in return period in mainstream and tributaries. Similar to the change in magnitude/intensity of the 100-year flood events under climate change impacts.

AC: The authors agree and can provide maps of the spatial variation of the change in return period. The maps will simply display the values of the presented violin plots of the changes in frequency and intensity. However, the authors consider placing these maps within in the supplement material to be able to display further extensive analysis on the dynamics in change drivers as suggested by the other reviewer.

7. The authors show that using hydro-SMILE provides a more robust extreme hydrological discharge values under climate change impacts, but do not discuss on how to make use of that finding in designing flood protection infrastructures, the problem that authors state from the beginning of the paper.

AC: Unfortunately, we the authors recede from suggesting designs for flood protection infrastructures since we lack expertise in engineering. However, we aim to provide the means for engineers to use the presented results for the design of such structures by stating that CC impacts should be considered in their design. The authors will add the following to the conclusions to make it clearer that this study does not cover designs but rather provides data to build upon for the design: "Further studies are necessary focusing on flood inundation to fully analyze the extent of the increase and frequency of this event for the design of flood protection infrastructure".

8. [Technical correction] Line 297, Page 12: "0.49 and 1.91 for 100 AM values (panel c) and 0.56 and 1.60 for 200 AM values (panel e)". Should it be (panel b) and (panel c)?

AC: The authors will change the mentioned values

---

## Author Response (AR1)

**Reply to Comments**

We want to thank all reviewers for their valuable and constructive comments!

Please find our point-by-point replies to the reviewer's comments below. The comments are taken from the original commentary. Replies are given in green below the respective comment.

**Reply to comments by reviewer 1:**

The authors added a new analysis (dynamics of driving mechanisms) to the manuscript and extended existing sections of the manuscript to clarify and describe the new additions (methods, results, discussion, and conclusion) as recommended by both reviewers. The supplemental material has also been updated and extended (model performance, dynamics of $HF_{100}$). Furthermore, we moved a part describing the benefit of the LE approach to the supplemental material. A point-by-point reply to the comments is given below.

1. The fourth paragraph in Introduction (Lines 73-77, Page 3) is vague. The two first sentences do not seem to be the reason for the choice of study area.

Reply: The authors rephrased this paragraph to clarify the choice of study area.

2. Can the authors explain the choices of CanESM2 and the use of RCP8.5 only?

Reply: The authors added a few explanations for the choice of a single GCM as well as a single scenario.

3. The study area has abundant in situ data. Should model parameters (i.e., soil properties) be calibrated?

Reply: The authors added a paragraph explaining the necessity for calibrating the hydrological model despite the abundance of data available for the region.

4. Since the reliability of the hydrological model affects the simulated discharge and the further analysis, the performance of the hydrological model should be presented and discussed in more detail. For example, for 16 gauges having NSE lower than 0.5 and 5 gauges having KGE lower than 0.5 (Lines 203-204, Page 7), an explanation is needed to show that the unsatisfactory is acceptable. For the other gauges with NSE and KGE higher than 0.5, how much higher than 0.5 are they? I think it is worth having maps that show the value of model performance metrics at all gauges.

Reply: The authors added further details about the model performance within the manuscript and added maps showing the spatial distribution of the model performance for each gauge within the supplementary material (S4).

5. Panels c, d, and e in Fig. 7 do not show the entire variation ranges.

Reply: The authors added the entire range of variation to the panels c, d, and e in Figure 7.

6. I think it is worth having a map that visualizes the spatial variation of the change in return period under climate change impacts, some interesting insights might be found. I am curious about the difference between the changes in return period in mainstream

and tributaries. Similar to the change in magnitude/intensity of the 100-year flood events under climate change impacts.

Reply: The authors added maps showing the spatial distribution of the changes in $HF_{100}$ frequency and magnitude to the supplementary material (S6) and refer the reader in the manuscript to this section of the supplements.

7. The authors show that using hydro-SMILE provides a more robust extreme hydrological discharge values under climate change impacts, but do not discuss on how to make use of that finding in designing flood protection infrastructures, the problem that authors state from the beginning of the paper.

Reply: The authors added the following statement to the end of the conclusion section of the manuscript to clarify that the authors are not able provide advice for the adaptation of flood protection infrastructures other than providing more robust values of future events: "Further studies are necessary focusing on flood inundation to fully analyze the extent of the increase and frequency of this event for the design of flood protection infrastructure".

8. [Technical correction] Line 297, Page 12: "0.49 and 1.91 for 100 AM values (panel c) and 0.56 and 1.60 for 200 AM values (panel e)". Should it be (panel b) and (panel c)?

Reply: The authors changed the references of the respective panels.

**Reply to comments by reviewer 2:**

The authors added a new analysis (dynamics of driving mechanisms) to the manuscript and extended existing sections of the manuscript to clarify and describe the new additions (methods, results, discussion, and conclusion) as recommended by both reviewers. The supplemental material has also been updated and extended (model performance, dynamics of $HF_{100}$). Furthermore, we moved a part describing the benefit of the LE approach to the supplemental material. A point-by-point reply to the comments is given below

1. The manuscript has a title of "…high return levels of peak flows…" but specifically focused on 100-year floods. With a large ensemble, the authors could investigate a range of high return levels of peak flows and see how the frequency, magnitude, and dynamics are projected to change in the future climate.

Reply: The authors added a statement to the manuscript explaining the reasons for focusing on the 100-year flood event to the manuscript.

2. The manuscript focused on proving that it is beneficial to have a large ensemble to estimate extreme peak flow events (Sections 3.1 and Figures 4 and 5), which, I think, is very obvious so I suggest moving them to the supplementary. The authors have generated a powerful dataset for extreme peak flow estimation, however, there is no analysis of the changes in flood frequency and magnitude. The authors are suggested to substantially expand the analysis on flood frequency and characteristics. See Yu et al. (2020) for an example of flood frequency analysis.

Yu, G., Wright, D. B., & Li, Z. (2020). The upper tail of precipitation in convection-permitting regional climate models and their utility in nonstationary rainfall and flood frequency analysis. *Earth's Future*, 8,

e2020EF001613. https://doi.org/10.1029/2020EF001613

Reply: The authors remained with their approach for a flood frequency analysis to illustrate the dynamics of the 100-year flood within a changing climate as provided within the manuscript in sections 2.2.4 and 3.2. We further added maps showing future changes in HF100 frequency and magnitude to the supplements (S6).

3. The authors only gave limited information on the evaluation results of the hydrologic model which are essential for building confidence in the following analysis. I suggest including figures and/or tables showing the evaluation results such as time series of flow events with other quantitative metrics such as correlation coefficient, % bias, root mean square error, KGE, NSE, etc. One related question would be how the level of trust (LOT) is calculated.

Reply: The authors further elaborated on the model's performance within the manuscript and referred the reader to other publications presenting the performance of the very same model. We also added a section to the supplemental material (S4) providing maps of the performance metrics.

4. The changing dynamics in the future climate (Section 3.2) are very interesting and worth digging into. The authors could dig into it with a mechanistic investigation of possible explanations for why they see such changes in future projections. For example, linking snow water equivalent and rain characteristics to the dynamical changes that are projected at the nivo-pluvial stations.

Reply: The authors added two new sub-sections (one in the method section and one in the result section) describing the methods to analyze the dynamics in driving mechanisms for extreme discharges above the $HF_{100}$ event and showing the results. We further added paragraphs to the discussion and conclusion sections regarding results of the dynamics in driving mechanisms.

Minor comments:

Lines 57-59: it is unclear how prediction is a reason for challenges in modeling and predicting high flows by Brunner et al. (2021a).

Reply: The Authors removed the prediction part from the manuscript.

Line 71: change "extraordinary" to "extreme".

Reply: The authors exchanged the term as recommended.

The paragraph starting from line 73: Needs substantial rephrasing. 1) Rephrase the sentence "This approach of high spatiotemporal resolution for climate and hydrological modeling is computationally demanding." Do the authors mean "This ensemble-based climate and hydrological modeling approach is computationally demanding because of the high spatio-temporal resolution"?

Reply: The authors rephrased the mentioned paragraph for better clarity.

2) Rephrase sentence "However, considering spatially refined catchment features (e.g., slopes, soil characteristics, land use), precise values due to higher temporal resolution, and the application of a SMILE for hydrological modeling supports an enhanced representation of extreme values within models." Do the authors mean high spatio-temporal resolution of hydro-SMILE is particularly valuable for an enhanced representation of extreme values in models because hydro-SMILE considers spatially-refined catchment features at high temporal resolutions?

Reply: The authors rephrased this paragraph for better clarity.

3) Rephrase "Thus, this study focuses on the major Bavarian river basins (upper Danube, Main, Inn) with all their tributaries". It is unclear to me what your study area has to do with the above two statements.

Reply: The authors rephrased this sentence to better explain that a high spatio-temporal resolution is beneficial and necessary for the heterogeneity within the study area.

Line 84: Remove "Therefore".

Reply: The authors removed 'Therefore' from the manuscript.

The paragraph that starts from Line 84: add section numbers throughout the paragraph.

Reply: The authors added section numbers as recommended.

Line 85: Confusing sentence. Remove "…to meet the requirements for the hydrological modeling."

Reply: The authors removed this part of the sentence as recommended.

Line 86: "…hydro-SMILE along..." should be "…along with…".

Reply: The authors added 'with' to this sentence as recommended.

Line 95: Remove "As a result"

Reply: The authors removed this phrase.

Line 102: "…(up to 1100 mm precipitation sums in the north, 2500 mm in the south; an average temperature of 10 °C in the north, down to 5 °C (-8 °C on alpine summits)…". Are the authors referring to annual total precipitation and annual mean temperature? Be clear on that. Also, be sure to mention the data sources for these numbers - are they from Poschlod et al. (2020) as well?

Reply: The authors rephrased this sentence to include the actual meaning of the values and added the reference 'Poschlod et al., (2020)' to the values within the brackets to better indicate their source.

Line 108: "The major river catchments were divided into a total of 98 smaller sub-catchments based on a common interest in flood protection and a more detailed variation in catchment characteristics, using a selection of gauges (Willkofer et al., 2020)." Rephrase this sentence. It is unclear to me whether the 98 sub-catchments were divided based on the spatial distribution of the 98 selected gauges or whether the gauges were selected because of the division of the 98 sub-catchments.

Reply: The authors rephrased the sentence to clarify the reason for selecting the 98 catchments.

Line 125: Figure caption of Figure 2: Also introduce what are SDCLIREF and WaSiM in the Figure caption.

Reply: The authors added the explanation for SDCLIREF and WaSiM to the Figure caption.

Line 142: What is "T63"?

Reply: T63 is a term describing the original grid resolution of the climate model. As it does not contribute to further understanding of the data, the authors removed this term.

Line 149: "Furthermore, the individual members of the CRCM5-LE are considered independent for the hydrological evaluation period from 1981 to 2099, as the analysis of variations in temperature and precipitation over land and ocean shows (Leduc et al., 2019)." This is the first time the authors mention "hydrological evaluation period". This is a confusing sentence. Aren't the CRCM5-LE individual members independent no matter what time period?

Reply: The authors combined this comment and the comment in line 214 and added the recommendation provided in the comment in line 214 here to clarify on the independence of the members of the CRCM5-LE.

Line 152: "…showing regional and seasonal variations in magnitude over Europe (Leduc et al., 2019)." What variables do the authors mean?

Reply: The authors will added the variables (temperature and precipitation) to this sentence in the manuscript.

Line 156: add "match" after "were adjusted to".

Reply: The authors added 'match' to this sentence.

Line 157: Change "RCM scale" to "RCM grid". Did the authors do the interpolation onto the RCM grid? If yes, be clear on what interpolation scheme is used.

Reply: The authors changed "RCM scale" to "RCM grid" and added the interpolation approach to the supplements (S3).

Line 160: "…3-hourly correction factors for every quantile and month". Unclear how the 3-hourly correction factor is applied.

Reply: The authors added a paragraph to the supplements explaining the application of the 3-hourly correction factors (S3).

Sentence starting on Line 162: I think the authors want to stress that bias correction is inevitable. Rephrase it to "Despite the benefits (increasing reliability of climate change projections of the hydrological impact model, reducing bias in mean annual discharge) and shortcomings (disrupting feedbacks between fluxes, modification of change signals, assumption of a stationary bias) of bias correction are highly debatable (e.g., Teutschbein and Seibert, 2012; Maraun, 2016; Ehret et al., 2012; Dettinger et al., 2004; Chen et al., 2021; Huang et al., 2014), bias correction is often inevitable for climate change impact studies (Gampe et al., 2019)."

Reply: The authors rephrased this sentence accordingly.

Line 168: For such a topographically complex region as described in the "Study area" section, I'm concerned the statistical downscaling between grids that are so different (from 12 km in RCMs to 500 m in hydrologic models) will lose important spatial heterogeneity across the domain. Does the mass-preserving approach address this problem?

Reply: The mass-preserving approach ensures that through the downscaling no additional precipitation is added or lost. Meaning, when the downscaled result is upscaled to the RCM grid the mass of the RCM grid is matched. The downscaling basically tries to distribute the coarse RCM information to a higher sub-grid scale. We don't think that the downscaling will alter the spatial heterogeneity. If so, then rather the bias correction will alter the spatial heterogeneity. As with all interpolation schemes, the obtained spatial result is only one of many possible spatial distributions.

Line 171: "The interpolation result was then applied to the SDCLIREF reference fields (Brunner et al., 2021b)". Unclear. Do the authors mean the SDCLIREF reference fields are also interpolated using the same method?

Reply: The authors added further information about the method to the manuscript.

Line 194: "minimizing a weighted combination of performance metrics, including Nash and Sutcliff efficiency (NSE; Nash 195 and Sutcliffe, 1970), Kling-Gupta efficiency (KGE; Gupta et al., 2009), the logarithmic NSE and the ratio of root mean squared error to standard deviation (RSR; Moriasi et al. (2007)) (Eq. (1))". Introducing the overall metric (OM) equation first and then describe what is in the equation. Then, give a threshold - what OM value is considered "good" or "bad"?

Reply: The authors introduced the overall metric first and then described it. We further provided an optimal value for the overall metric and clarified the meaning of deviations from that optimum.

Line 203: "(NSE: 16; KGE: 5)" Unclear. What does this mean?

Reply: The authors rephrased this sentence to better clarify the meaning of these numbers.

Line 208: "Consequently, the level of trust (LOT) for peak flows of return periods of 5, 10, and 20 years of flood events, introduced in Willkofer et al. (2020) showed a moderate to high confindence for most catchments, with gauges of poor simulated performance yielding a

lower LOT with increasing return levels." Do the authors mean gauges with good performance have higher LOT for peak flows with return periods of 5, 10, and 20 years, whereas gauges with poor performance have lower LOT, especially for peak flows at longer return periods?

Reply: The authors rephrased this sentence to clarify the meaning of the LOT and added a description to the supplemental material as well.

Line 214: "The entire modeling period is shortened by ten years to account for the time span it takes the RCM to produce fully independent realizations due to the inertia of the ocean model (Leduc et al., 2019)." I have two comments: 1) I saw that the authors partially address my question for Line 149 here. It would be better to rearrange this part and the sentence on Line 149 such that the 10-year spin-up period and the choice of the evaluation time period are more clearly lined up and explained. 2) Rephrase this sentence to "We focus on 1961—2099 as opposed to 1950 – 2099 to account for the time it takes for the RCM to produce fully independent realizations due to the inertia of the ocean model (Leduc et al., 2019)."

Reply: As mentioned in the reply to comment of line 149, the authors rephrased the sentence and moved it from line 214 to 149 to introduce the independence of the different members earlier in the manuscript.

Figure 3: What is "HF T,BM" on the far right?

Reply: The authors added the explanation for this term within the figure caption of the manuscript.

Line 256 and thereafter: Change "intensity" to magnitude throughout the manuscript.

Reply: The authors changed the term throughout the manuscript.

Line 293: "…as indicated by the spread of the blue markers around the black benchmark line." I think the authors mean "… as indicated by the decreasing spread of the blue markers around the benchmark line with increasing sample size."

Reply: The authors changed this phrase accordingly.

Lines 296-297: What are panels a, c, and e?

Reply: This is a clear mistake in referencing the different parts of the Figure. The authors changed the references to Fig. 4a, 4b, and 4c.

Figures 1&6: What do "balanced" and "unbalanced" pluvial mean?

Reply: The authors added a short explanation of these terms to the respective location within the manuscript.

---

## Author Response (AR2)

**Response to reviewer comments**

Based on observation data from 98 hydrological stations, the author first systematically evaluated the superiority of the hydro SMILES model in simulating high flow (100 year flood) during the large recurrence period, and then estimated the frequency and intensity changes of 100 year flood in the near, medium, and long term in the future. This is a revised manuscript, and I believe it has reached the level of HESS in terms of expression, organization, technical method description, and innovative points. This manuscript will also attract widespread attention from scholars related to hydrological simulation, climate change impacts, and flood prediction. However, the following issues need to be considered before publishing the manuscript.

AC: The authors would like to thank the reviewer for the valuable comments to further improve the quality of the manuscript. Detailed responses (author comments AC) are provided for each specific major and minor comment (RC) below.

**Major comments:**

Since the emphasis is on the benefits of hydro-SMILEs model for simulating high flows with extreme return periods, a comparison analysis with results from existing models is ofter required. Although the authors did not conduct a comparative analysis with other models, it is also necessary to clear the so-called more appropriate or beneficial based on the results of existing studies. The current discussion is too superficial, it is suggested to add some comparative discussions.

AC: The authors added a comparative statement to the first paragraph of the discussion. It compares our study to the results by van der Wiel et al. (2019) who used a slightly different approach for the creation of the large ensemble of climate model data to drive a global hydrological model to create a large ensemble of modelled discharge to show the benefit of a large ensemble for the estimation of extreme return levels of discharge. We further added a statement of how our approach is beneficial as it reduces uncertainties originating from different extreme value distributions as shown by Lawrence (2020). We further added a comparative reference for the opportunity to analyse dynamics of extreme high flows and their driving mechanisms (Brunner et al., 2021).

**Minor comments:**

There are some abbreviations of terminology in the manuscript. It is recommended to retain abbreviations of nouns used multiple times in the abstract, and implement abbreviations of nouns used only once when they first appear in the main text (such as GEV). In addition, some abbreviations (such as Line 16 and Line 66, SMILE) have already been presented in the abstract and do not need to be repeated in the main text.

AC: The authors scanned the manuscript for repetitive definitions of abbreviations and removed them within the text. If an abbreviation is defined within the text and a figure caption, we decided to maintain these definitions as we think it provides a quicker understanding of the terms mentioned within the figure for the reader.

**Lines 52-53 and 57**. Please first cite Brunner et al., 2021a, followed by Brunner et al., 2021b. There is no problem with the literature cited by the author, but the order in which a and b appear needs to be adjusted.

AC: The authors thank the reviewer for this comment. However, the order of appearance of these specific references is bound to the alphabetical order of the authors of the respective publications which determines their order within the reference list as well as the citation style which is a HESS requirement. Furthermore, we found that this odd appearance of references also occurs within other HESS publications (see Brunner et al., 2019 citing Brunner et al. 2019b before Brunner et al. 2019a; or Höge et al., 2022 citing Kratzert et al., 2019b before Kratzert et al., 2019a). Thus, although we agree that the order of appearance is unusual, we cannot change this order in an automated manner. However, if the reviewer insists to have this order changed, the authors will change it manually throughout the manuscript. However, it might as well be changed during copy editing.

**Fig.3**. Replace 'nxm' to 'n×m', 'nxHFT,m' to 'n×HFT,m'.

AC: The authors changed the multiplication character within the entire Figure 3 according to the recommendation.

**Lines 23 and 293**. Repeating abbreviations.

AC: The authors removed the repeating abbreviations here.

**Line 169**. Provide reference for the mass preserving approach.

AC: The authors added references for the mass preserving approach.

**Lines 315-316**. Provide the basis for dividing into three time periods.

AC: Added the basis for the division into the three mentioned future periods.

**Line 334**. Provide reference for the MK.

AC: The authors provided references for the Mann-Kendall test.

**Line 449-451**. Repetitive statements.

AC: The authors removed the last sentence of the paragraph as its statement is repeated by the first sentence of the following paragraph.

References

Brunner, M. I., Farinotti, D., Zekollari, H., Huss, M., and Zappa, M.: Future shifts in extreme flow regimes in Alpine regions, Hydrol. Earth Syst. Sci., 23, 4471–4489, doi:10.5194/hess-23-4471-2019, 2019.

Brunner, M. I., Swain, D. L., Wood, R. R., Willkofer, F., Done, J. M., Gilleland, E., and Ludwig, R.: An extremeness threshold determines the regional response of floods to changes in rainfall extremes, Commun Earth Environ, 2, doi:10.1038/s43247-021-00248-x, 2021.

Höge, M., Scheidegger, A., Baity-Jesi, M., Albert, C., and Fenicia, F.: Improving hydrologic models for predictions and process understanding using neural ODEs, Hydrol. Earth Syst. Sci., 26, 5085–5102, doi:10.5194/hess-26-5085-2022, 2022.

Lawrence, D.: Uncertainty introduced by flood frequency analysis in projections for changes in flood magnitudes under a future climate in Norway, Journal of Hydrology: Regional Studies, 28, 100675, doi:10.1016/j.ejrh.2020.100675, 2020.

van der Wiel, K., Wanders, N., Selten, F. M., and Bierkens, M. F. P.: Added Value of Large Ensemble Simulations for Assessing Extreme River Discharge in a 2 °C Warmer World, Geophys. Res. Lett., 46, 2093–2102, doi:10.1029/2019GL081967, 2019.